# Engineered CRISPRa enables programmable eukaryote-like gene activation in bacteria

Yang Liu[1,2], Xinyi Wan[1,2] & Baojun Wang [1,2]

Transcriptional regulation by nuclease-deficient CRISPR/Cas is a popular and valuable tool for routine control of gene expression. CRISPR interference in bacteria can be reliably achieved with high efficiencies. Yet, options for CRISPR activation (CRISPRa) remained limited in flexibility and activity because they relied on $\sigma^{70}$ promoters. Here we report a eukaryote-like bacterial CRISPRa system based on $\sigma^{54}$-dependent promoters, which supports long distance, and hence multi-input regulation with high dynamic ranges. Our CRISPRa device can activate $\sigma^{54}$-dependent promoters with biotechnology relevance in non-model bacteria. It also supports orthogonal gene regulation on multiple levels. Combining our CRISPRa with dxCas9 further expands flexibility in DNA targeting, and boosts dynamic ranges into regimes that enable construction of cascaded CRISPRa circuits. Application-wise, we construct a reusable scanning platform for readily optimizing metabolic pathways without library reconstructions. This eukaryote-like CRISPRa system is therefore a powerful and versatile synthetic biology tool for diverse research and industrial applications.

[1] School of Biological Sciences, University of Edinburgh, Edinburgh EH9 3FF, UK. [2] Centre for Synthetic and Systems Biology, University of Edinburgh, Edinburgh EH9 3FF, UK. Correspondence and requests for materials should be addressed to B.W. (email: baojun.wang@ed.ac.uk)

I n recent years, reprogrammable CRISPR (Clustered Regularly Interspaced Short Palindromic Repeats)-mediated genetic regulation has emerged as a powerful tool for synthetic biology research. Transcriptional regulations through CRISPR interference (CRISPRi) and activation (CRISPRa) were widely used to modulate gene expression[1–10]. Bacterial CRISPRi is a matured technology with plenty of examples but for bacterial CRISPRa, few have been demonstrated[2,10]. The established mechanisms have RNA polymerase-recruiting domains linked to a dCas9–sgRNA complex, and rely on direct interactions between the complex and a σ[70] promoter. These designs suffered from low dynamic ranges of activation output and could not support multi-input activation because the operator must lie near the core promoter regions, and the σ[70] promoter per se must be weak enough for activation to be observed[10].

In contrast, σ[54]-dependent promoters were much overlooked potential targets for CRISPRa. A σ[54] factor "locks" a σ[54]-dependent promoter in a stable closed complex that tightly blocks transcription initiation[11,12]. A σ[54]-activator binds to an upstream activating sequence (UAS) and performs long-distance regulation through a local DNA loop, and catalyzes ATP hydrolysis to unlock the complex for transcription initiation. This mechanism is similar to transcriptional activation involving the eukaryotic RNA polymerase II in several ways and is known as "eukaryote-like" regulation in bacteria[13–18]. Hence, σ[54]-activators are often termed "bacterial Enhancer Binding Proteins" (bEBPs)[18,19].

Here, we construct and characterize a CRISPRa system based on a eukaryote-like activation mechanism in bacteria. This CRISPRa device shows strong activity, superior dynamic range (ON–OFF dynamic range > 1000 and > 70-fold between cognate and mismatch sgRNA) and remarkable tolerance to a wide range of UAS locations. The device can work in both *Escherichia coli* and non-model bacteria, and can activate multiple high value wild-type (WT) σ[54]-dependent promoters. Use of dxCas9 enables non-canonical PAM targeting and further improves the dynamic ranges of the system to practical application-required levels. Finally, we set up an efficient profile scanning platform for metabolic pathways, in which the sgRNA library and thus its encoded transcriptional activation profiles do not need to be rebuilt and can be applied to different pathways. Those profiles, when applied to multi-gene expression, show good stability and durability. This CRISPRa device not only enriches the genetic regulation toolbox, but also supports real-life application scenarios by its high performance.

## Results

### Design of eukaryote-like CRISPR activation in bacteria.
To build a eukaryote-like programmable activation system, our first step was to identify a functional σ[54]-activation domain. It has been proven that the bEBP PspF from *E. coli* has a highly modular structure—a truncated PspF remained as a functional activation domain in a three-hybrid system in vitro[20,21]. A similar strategy was adopted to bridge the sgRNA and the activator—an RNA-binding peptide λN22plus was fused to the PspF activation domain (PspFΔHTH) at the C-terminus to create PspFΔHTH:: λN22plus. We designed the sgRNA scaffold to incorporate two BoxB RNA aptamers, which recruit λN22plus peptide fused proteins, at the Tetra-loop and the stem loop 2 (Fig. 1b)[22,23]. To optimize sgRNA scaffold, we also mutated the U-A pair to G-C at "+5"[23]. We then set up a test platform using the σ[54]-dependent P$_{pspA}$ promoter. In total, 3 bp downstream of the native UAS were mutated to an NGG PAM, which could be targeted by a dCas9–sgRNA complex. A sfGFP reporter was placed downstream to monitor the transcriptional output (Fig. 1a). We verified that this promoter could still be activated by its cognate

activator PspF in *E. coli* MC1061ΔpspF (Fig. 1e). These components, when supplemented by dCas9 and transformed into *E. coli*, successfully generated a transcriptional activation response similar to that of wild-type PspF, demonstrating the proof-of-concept of our CRISPRa system on σ[54]-dependent promoters (Fig. 1e). We then generated another UAS-gRNA set by replacing the UAS of P$_{pspA}$ by two copies of UAS (G6) from the P$_{hrpL}$ promoter and modified the sgRNA correspondingly. A combinatorial activation test showed that the σ[54]-dependent promoters could only be activated when the UAS sequence matched the sequence of the spacer region in the sgRNA (Fig. 1e), proving that our CRISPRa system critically relied on sgRNA-UAS specificity and had little cross-talk issues.

Given that many CRISPRa systems in eukaryotes were built by fusing the activation domain to dCas9 directly[3,5], we asked if the same could be applied to the PspF activation domain. The HTH domain truncated PspF was fused to dCas9 and CRISPRa was attempted, but was unsuccessful (Supplementary Fig. 1c). This result was within expectation given that PspF must assemble as a hexamer to function, and dCas9 probably interfered with the assembly process as it was relatively bulky[24,25].

To demonstrate the necessity of dCas9, sgRNA, PspFΔHTH:: λN22plus (or simply, activator) and the σ[54] factor (RpoN) in our CRISPRa system, we constructed reporter circuits with or without the dCas9 generator, the sgRNA generator and the activator generator circuits on three different plasmids. These plasmids were combinatorially mixed with empty vector controls and co-transformed into cells. A strong response could only be achieved when all three components were present, with a dynamic range of 78-fold between the cognate and the mismatched sgRNA (Fig. 1c). Therefore, our CRISPRa activity can be controlled and tuned on three dimensions. This CRISPRa device was non-functional in a ΔrpoN strain and thus is σ[54]-dependent (Supplementary Fig. 1e). Regarding the requirement of hexamer assembly of the activator, we further asked if only one UAS would suffice to stabilize hexamer assembly, by providing only two instead of four RNA aptamers that act as local docking sites for activators. Our results show that a single UAS remains functional in its activity (Supplementary Fig. 1a).

One of our goals was to create a CRISPRa system that does not depend on the host strain, so the CRISPRa that successfully worked in MC1061ΔpspF was transformed into and tested in a general cloning strain *E. coli* TOP10. The dynamic range was similar to that in MC1061ΔpspF but the maximum activity was halved (Supplementary Fig. 2). We reasoned that it might be due to potential background expression of the endogenous PspA protein controlled by P$_{pspA}$ promoter on the genome, which is known to allosterically inhibit PspF[20,26]. This hypothesis was indirectly supported by the observation that expression of PspFΔHTH::λN22plus (without sgRNA for CRISPRa) led to a higher activation efficiency on wild-type P$_{pspA}$ in TOP10 than in MC1061ΔpspF (Supplementary Fig. 1f). As the engineered PspFΔHTH::λN22plus has no DNA binding domain to target P$_{pspA}$ on the genome, it was likely that the endogenous PspA expression was activated by heterohexamers comprising PspFΔHTH::λN22plus and PspF and increased levels of PspA in turn impaired the efficiency of the activator on our engineered promoter in TOP10. It is worthwhile to mention that the lower output level in TOP10 may also be caused by other unknown mechanisms.

Nonetheless, our CRISPRa system still retained significant activity for general purposes. The rest of this study, except for those in Fig. 3d (see below), were done in the strain MC1061ΔpspF as it provided a higher maximum output so small changes in activities could be readily detected. In addition, to simplify our system for further optimizations, we put the

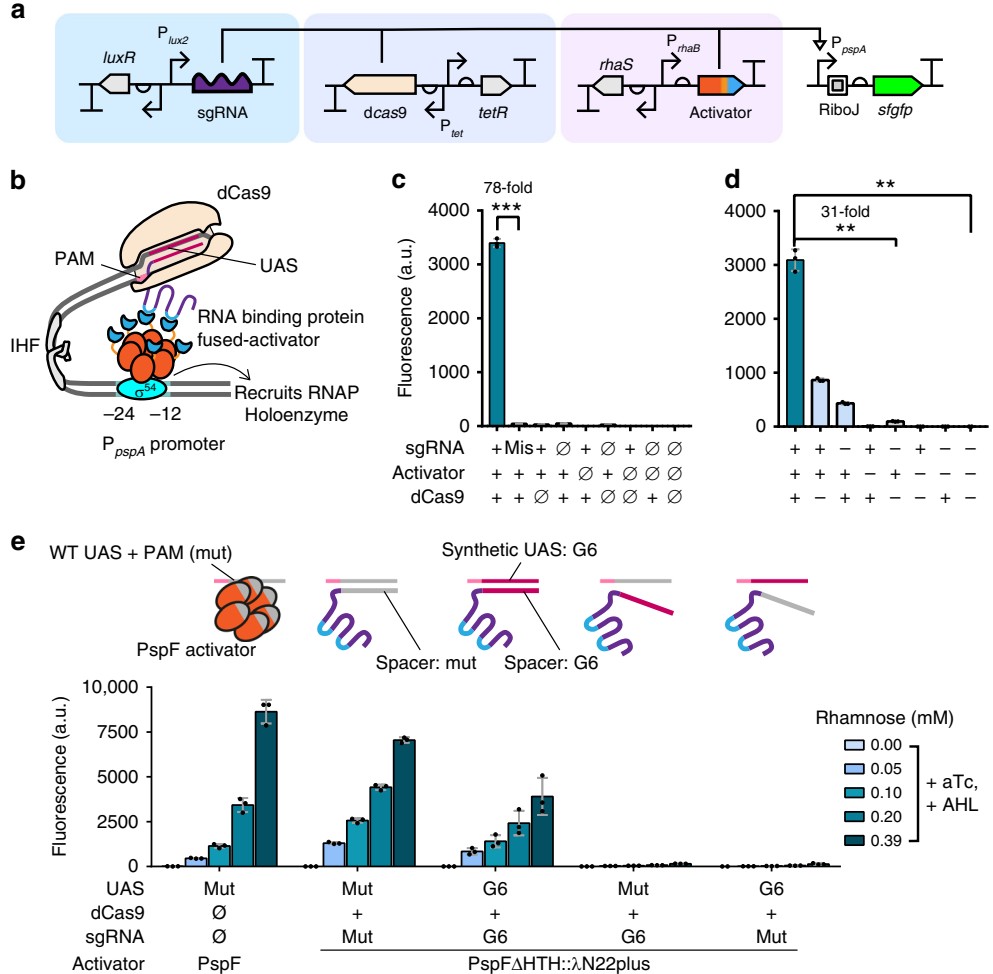

**Fig. 1** Design and function of eukaryote-like CRISPR activation in *E. coli*. **a** The circuit design and structure of the CRISPRa device. The circuit includes a sgRNA generator driven by $P_{lux2}$ (light blue block, inducible by AHL), a dCas9 generator driven by $P_{tet}$ (blue block, inducible by aTc), an activator generator driven by $P_{rhaB}$ (red block, inducible by rhamnose), and a sfGFP reporter driven by $P_{pspA}$ with wild-type or heterologous UAS. **b** The activation mechanism of a eukaryote-like long distance regulation in our CRISPRa design, based on the IHF-dependent DNA loop structure. **c**, **d** Test of necessity of the three components in CRISPRa. **c** Combinations of components were achieved via presence (+) or absence (∅) of genetic part in the strain. The mismatch sgRNA had a random sequence as its spacer (sgRNA-LEA3). All strains were cultured with AHL (1.6 μM), rhamnose (0.4 mM) and aTc (2.5 ng mL$^{-1}$). Statistical difference was determined by a two-tailed Welch's t-test: (sgRNA cognate/mismatch) $p = 0.0002$, $t = 74.98$. **d** All genetic components for CRISPRa were present in the strain and combinations were achieved via presence (+) or absence (−) of inducers. Inducer concentrations: AHL (1.6 μM), rhamnose (0.4 mM) and aTc (2.5 ng mL$^{-1}$). Statistical difference was determined by a two-tailed Welch's t test: (+++/−−−), $p = 0.0014$, $t = 26.82$, (+++/−+−), $p = 0.0015$, $t = 25.98$. **e** A PAM inserted $P_{pspA}$ with wild-type UAS (referred as mut) or heterologous UAS (referred as G6) was tested against different spacers for activation in a *pspF* knocked out strain *E. coli* MC1061Δ*pspF*. Concentrations of inducers: aTc (2.5 ng mL$^{-1}$), AHL (1.6 μM). Error bars, s.d. (*n* = 3); a.u., arbitrary units; *p* value summary: ****$p$ value < 0.0001, 0.0001 < ***$p$ value < 0.001, 0.001 < **$p$ value < 0.01, 0.01 < *$p$ value < 0.05, $p$ value ≥ 0.05: n.s. Source data are provided as a Source Data file

activator under the control of a constitutive Anderson promoter BBa_J23106[27], which provided the best output function compared to three other promoters of different strengths tested (Supplementary Fig. 1b).

**Optimization of target sites and sgRNA based dynamic ranges.** Successful activation of the σ54-dependent promoter critically depends on its local DNA loop structure, which establishes a two-dimensional spatial relationship between the UAS and the transcription start site. This means that the spatial direction of the engineered activator relative to the axis of the DNA helix may affect the efficiency of our CRISPRa system (Fig. 2a). To understand the relationship between UAS positions and the spatial directions of the engineered PspF hexamer, we designed a library of $P_{pspA}$ where the synthetic UAS was progressively shifted

upstream from its original location (defined as "0"), and measured the resulting efficacies of CRISPRa. CRISPRa remained functional within a shift window of 40 bp. Most interestingly, the activities alternated between crests and troughs spaced at 5 bp apart. This oscillation exactly matches the fact that the DNA helix completes a turn every 10 bp (Fig. 2a), and strongly suggested a phase-dependency between the UAS and the core promoter. We thus sought further evidences and augmented the library to zoom into the 0–10 bp region corresponding to half a wave length, and indeed observed a sinusoidal waveform. This might provide insights on the underlying molecular mechanism of our CRISPRa system.

In our CRISPRa output characterization, the metric dynamic range could be defined in different ways depending on what constitutes the OFF state. Our CRISPRa system achieved >1000-fold dynamic range for all component induced versus

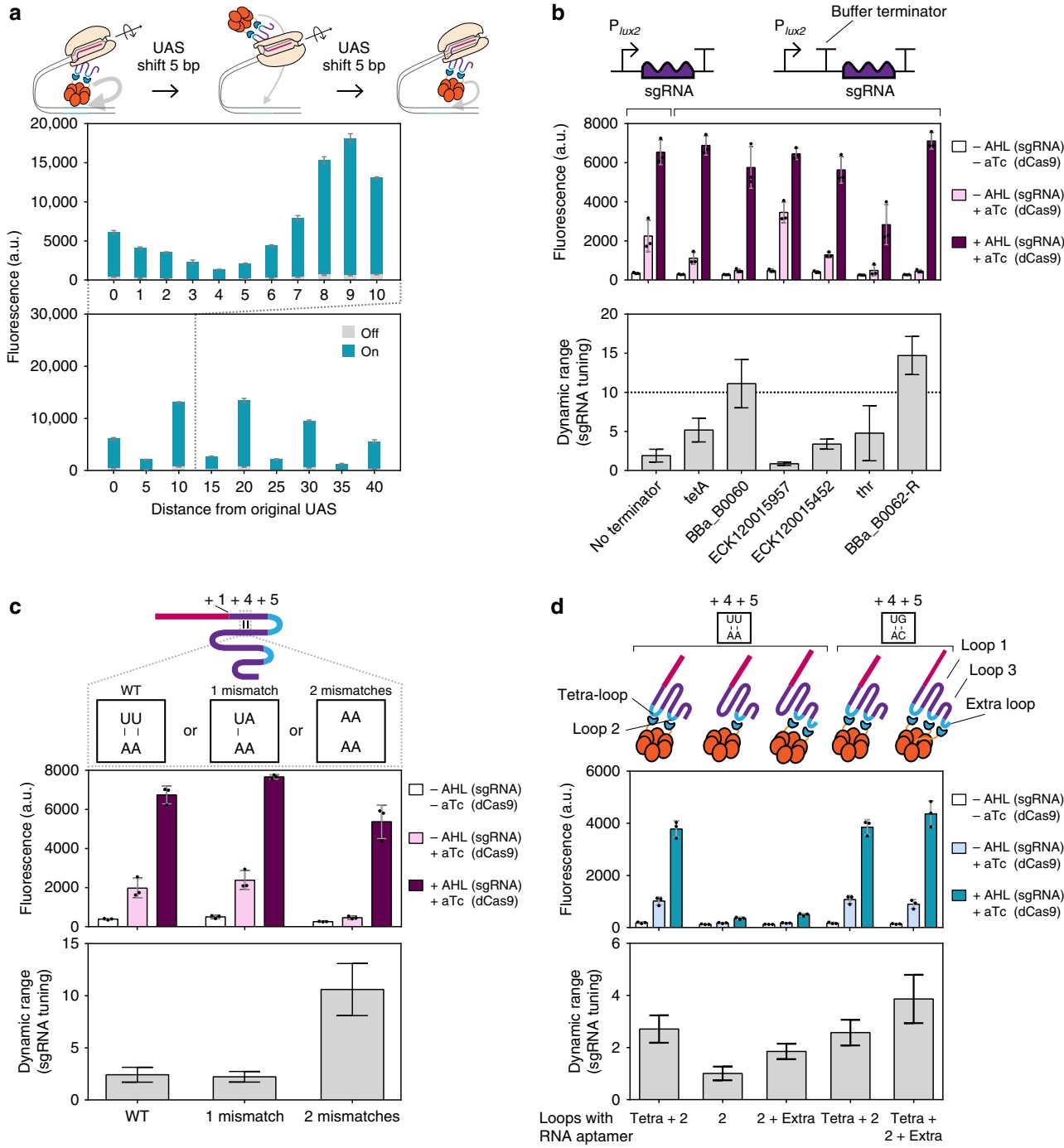

**Fig. 2** Optimization on synthetic UAS sites and sgRNA-dependent dynamic ranges. **a** Spatial relationship between the activator and the core P$_{pspA}$ with an artificial UAS LEB2 (Supplementary Data 1). Activator was driven by the constitutive promoter BBa_J23106. Concentrations of inducers used in the ON state: 0.4 μM AHL (sgRNA) and aTc 2.5 ng mL$^{-1}$ (dCas9). No inducers were added in the OFF state. **b** sgRNA-dependent dynamic range optimization by "Buffer Terminators" which were previously characterized[59]. **c** sgRNA-dependent dynamic range optimization by mismatches on the sgRNA scaffold. **d** sgRNA-dependent dynamic range optimization by moving RNA aptamer to different positions. The first three designs contained a wild-type U-A pair at the "+5" site. The circuits for **b**–**d** all employed P$_{pspA}$ with two synthetic UAS LEA2 (Supplementary Data 1) and the activator was driven by the constitutive promoter BBa_J23106. Concentrations of inducers used: 0.4 μM AHL (sgRNA) and aTc 2.5 ng mL$^{-1}$ (dCas9). Error bars, s.d. ($n = 3$); a.u., arbitrary units. Source data are provided as a Source Data file

non-induced (Supplementary Fig. 3a), > 70-fold for cognate versus mismatch sgRNA (Fig. 1c), and > 30-fold in the scenario where dCas9 and sgRNA were treated as a single inducible unit (Fig. 1d). These values surpassed the highest records to date[10]. However, these values are not practically relevant when CRISPRa needs to be applied to multi-gene regulation and transcriptional

cascades, where dCas9 and the activator should be constantly available at a fixed level and the sgRNA circuits need to be pre-installed. Under this scenario, dynamic ranges are strictly defined by circuit outputs between sgRNA induced and sgRNA uninduced states. Our results indicated that leaky sgRNA transcription owing to a leaky inducible promoter activation

explained the significant basal activation observed in the inducible device (Fig. 1c, d). To optimize the CRISPRa system and improve its dynamic ranges (defined by induced versus non-induced sgRNA), we first investigated its input–output relationship, and discovered that low induction levels for sgRNA already led to saturated CRISPRa responses (Supplementary Fig. 4). Therefore, we reasoned that a decrease in the range of sgRNA input, either by reducing sgRNA induction efficiencies or by reducing sgRNA's affinities to dCas9, would decrease the leakiness of CRISPRa, whereas excessive sgRNA would maintain its maximum output. To this end, we explored three separate strategies in optimizing sgRNA-dependent dynamic ranges. First, we inserted various known terminators as buffer sequence (named as "buffer terminators") between $P_{lux2}$ and sgRNA to interfere the function of the promoter and the output ranges of sgRNA transcription. Some of them reduced sgRNA leakiness and significantly improved the dynamic ranges to >10-fold (Fig. 2b). Second, we employed a wild-type U-A pair at the "+5" site and introduced no, one or two nucleotide mismatches into the original sgRNA scaffold, and hypothesized a reduction in the resulting dCas9–sgRNA affinity. sgRNA with two mismatching nucleotides could increase the dynamic range significantly to ~ 10-fold (Fig. 2c). Third, we moved the RNA aptamers onto different positions of the sgRNA scaffold, but discovered that our initial design was both necessary and sufficient (Fig. 2d). We further applied our first and second strategies to different spacers and the "mismatch in sgRNA strategy" was generally more versatile (Supplementary Fig. 5).

**CRISPRa is functional on different σ54-dependent promoters.** Our success in swapping the wild-type UAS in mutant $P_{pspA}$ with an orthogonal UAS suggested that our CRISPRa system could be repurposed to activate any σ54-dependent promoters, provided that a PAM could be found near the native UAS. This would be immensely useful for studying σ54-dependent promoters, because unlike PspF, many σ54-activators from bacteria are response regulators for environmental stimuli[28]. Thus, activation of their cognate promoters could be hardly achieved without changing the culture condition, which inevitably introduces additional factors in an experiment.

To demonstrate the function of our CRISPRa system on σ54-dependent promoters, we identified and cloned several promoters with research relevance: The wild-type $P_{pspA}$ promoter was involved in phage shock response. $P_{hrpL}$ from *Pseudomonas syringae* regulates Type III Secretion System-mediated pathogenicity in plants[29], and promoters $P_{nifH}$ and $P_{nifJ}$ control a nitrogen fixation pathway in *Klebsiella oxytoca* that responds to nitrogen and oxygen availabilities[30]. For these promoters, we intended to maintain sequence integrity, so we opted for PAM sites near the wild-type UAS instead of inserting new ones. All promoters could be successfully activated by our CRISPRa system to drive sfGFP expression (Fig. 3a). We further asked if our CRISPRa system could activate endogenous σ54-dependent promoters on *E. coli* genome, rather than just episomes. We targeted *pspA* in MC1061Δ*pspF* for activation and employed RT-qPCR to measure the relative transcription activities from $P_{pspA}$ and $P_{norVW}$, and results showed that the former was activated in a specific manner (Supplementary Fig. 6).

We also demonstrated that the CRISPRa systems with engineered PspFΔHTH::λN22plus built for $P_{nifH}$ and $P_{nifJ}$ were functional when transferred to the original host species *K. oxytoca* (Fig. 3b), proving that our CRISPRa system could be heterologously expressed to activate σ54-dependent promoters in other species. This should be helpful for bypassing their natural genetic regulatory networks[31].

It is worth mentioning that the relative activation efficiencies on wild-type σ54-dependent promoters by the CRISPRa system versus efficiencies mediated by their wild-type cognate activators has no fixed value even when the transcription levels of the activator genes were similar (Supplementary Fig. 7). This might be affected by the differences in targeting affinities between the CRISPRa complexes and their wild-type cognate activators and suboptimal UAS sequences owing to limitations in the availability of NGG PAM positions on the wild-type promoters.

As an example of utilizing this CRISPRa as a tool for σ54-dependent promoter related research, we employed our CRISPRa system to research potential modularity of the DNA components within a σ54-dependent promoter. σ54-dependent promoters are diverse in structures, with different UAS (in strength and numbers), loop regions, and core sequences[32,33]. Based on consensus sequences, we standardized, in length, three DNA loop regions and three core sequences from different σ54-dependent promoters. We then connected them combinatorially, with identical linker DNA in between, and placed them downstream of a synthetic UAS (Fig. 3c). Our results showed that except for the $P_{flhDp2}$ core sequence, DNA loop regions and core sequences are modular and could be shuffled to yield functional hybrid promoters (Supplementary Fig. 8). In addition, the relative ranks of dynamic ranges conferred by one core sequence is consistent when paired with different loop regions. This is also true when the same loop region is pair with different core sequences. So the $P_{hrpL}$ loop region and the $P_{pspA}$ core sequence, which were consistently the strongest when paired with different elements, could combine to give a hybrid promoter with the highest dynamic range (Fig. 3c).

**Alternative σ54-activator mining by protein engineering.** To minimize the influence of the host strain on our CRISPRa system owing to potential PspFΔHTH::λN22plus/PspF heterohexamers formation, we searched for other σ54-activators, which might be orthogonal to the PspF-based activator. This however posed a challenge, because many bEBPs contain a sensory module domain at their N-termini, which must be removed to avoid interference from their regulatory pathways in the host[28].

One such example is the bEBP NorR from *E. coli*, involved in NO sensing[34,35]. NorR with its sensory module deleted (NorRΔ-GAF) strongly activates its cognate promoter $P_{norV}$ in absence of NO[36,37]. We thus replaced the HTH domain in NorRΔGAF by λN22plus as we did for PspF and tested its activation function, but no significant activity was observed. Previous researches indicate that for two bEBPs, HrpS, and HrpR, the HTH domain may be required for assembly into the HrpRS hexamer[29]. We conjectured that the same applied to NorR, and so generated two constructs with most of the NorRΔGAF fused directly to the λN22plus peptide, which successfully yielded activities in CRISPRa (Fig. 3d). Subsequently, we tested three other bEBPs using the same approach, and identified a new functional activator based on WtsA, which regulates expression of pathogenic genes in *Erwinia stewartii*[38] (Fig. 3d, Supplementary Fig. 10). Unlike NorRΔGAF::λN22plus, WtsA::λN22plus did not impose significant metabolic burden onto *E. coli* (Supplementary Fig. 9).

**Orthogonality based on gRNA and UAS.** Orthogonality is a prerequisite for multiplex gene regulation[39,40]. Our CRISPRa comprises multiple components and thus offered opportunities to engineer orthogonality on different levels. We first capitalized on the specificity between UAS and gRNA (Fig. 1e) to create a library of orthogonal UAS-gRNA pairs. We designed a library of synthetic $P_{pspA}$ and generated unique UAS by sequence randomization. Based on the functional window for UAS locations (Fig. 2a),

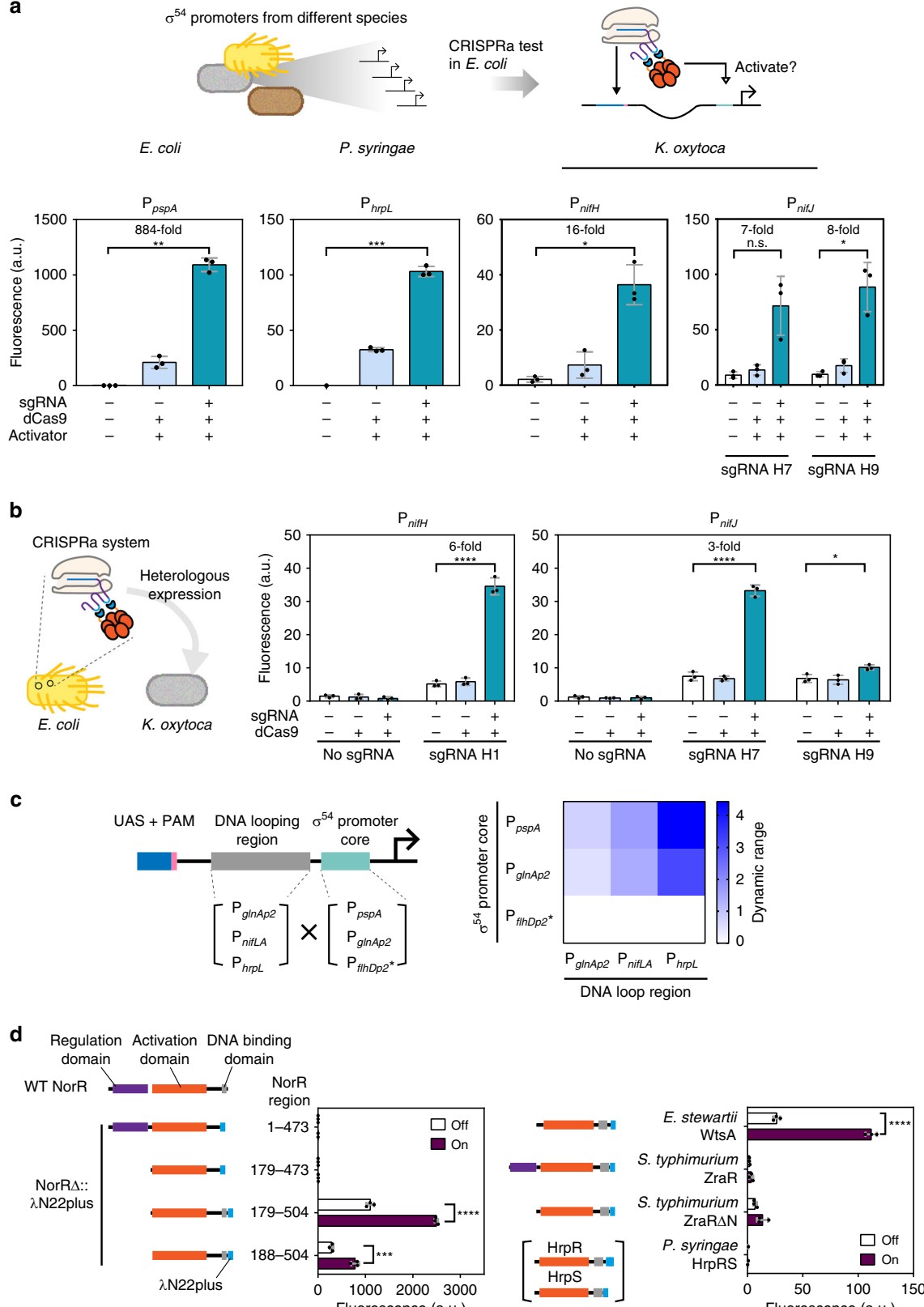

we designed two synthetic enhancers with PAM at −108 to −110 (LEB) and at −128 to −130 (LEA) respectively. The 10 sgRNA generators with the 5 CRISPRa circuits carrying various UAS were co-transformed and tested for activation. The results confirmed that the synthetic $P_{pspA}$ could only be activated by its cognate sgRNA (Fig. 4a). This high orthogonality will allow

construction of multiplex and layered CRISPRa with little crosstalks. We then measured orthogonality between two RNA aptamer–adaptor pairs, namely, BoxB-λN22plus against MS2-MCP, in triggering CRISPRa using the same design. Again, significant orthogonality was observed between the aptamer–adaptor pairs (Fig. 4b, Supplementary Fig. 11).

**Fig. 3** Functional expansion with different σ54-dependent promoters and bEBPs. **a** Activation of P$_{pspA}$, P$_{hrpL}$, P$_{nifH}$, P$_{nifJ}$ in *E. coli*. Inducer concentrations used: For P$_{pspA}$ and P$_{hrpL}$ activation, 1.6 μM AHL (sgRNA), 2.5 ng mL$^{-1}$ aTc (dCas9), 0.4 mM rhamnose (activator). For P$_{nifH}$ and P$_{nifJ}$ activation, 1.6 μM AHL (sgRNA), 2.5 ng mL$^{-1}$ aTc (dCas9), 3.2 mM rhamnose (activator). Statistical difference was determined by a two-tailed Welch's *t* test: P$_{pspA}$, $p = 0.0011$, $t = 30.69$; P$_{hrpL}$, $p = 0.0006$, $t = 38.34$; P$_{nifH}$ (H1), $p = 0.0134$, $t = 8.093$; P$_{nifJ}$ (H7), $p = 0.0536$, $t = 4.053$; P$_{nifJ}$ (H9), $p = 0.0242$, $t = 6.137$. The data of P$_{nifH}$ and P$_{nifJ}$ are identical to those shown in Fig. 4c. **b** P$_{nifH}$ and P$_{nifJ}$ activation in *K. oxytoca*. Inducer concentrations: 2.5 ng mL$^{-1}$ aTc (dCas9), 3.2 μM AHL (sgRNA). Statistical difference was determined by a two-tailed *t* test: P$_{nifH}$ (H1), $p < 0.0001$, $t = 18.85$; P$_{nifJ}$ (H7), $p < 0.0001$, $t = 20.73$; P$_{nifJ}$ (H9), $p = 0.0165$, $t = 3.976$. **c** The dynamic ranges of hybrid σ54-dependent promoter activation. The asterisk represents the P$_{flhDp2}$ was a predicted σ54-dependent promoter[60]. Activator expression was driven by the constitutive promoter BBa_J23106. Inducer concentrations used: 1.6 μM AHL (sgRNA), 2.5 ng mL$^{-1}$ aTc (dCas9). **d** Orthogonal activators in *E. coli* TOP10. Left: Truncated NorRs were fused with a λN22plus adaptor. Right: four activators without the HTH domain truncations were fused with a λN22plus adaptor and tested. All engineered activators were driven by the promoter BBa_J23106. P$_{pspA}$-20 was used for the reporter, which could be activated by sgRNA-LEB2. Inducer concentrations used: 0.025 μM AHL (sgRNA), 2.5 ng mL$^{-1}$ aTc (dCas9) for ON states. Statistical difference was determined by a two-tailed *t* test: NorR (179-504), $p < 0.0001$, $t = 27.54$; NorR (188–504), $p = 0.0007$, $t = 9.456$; WtsA, $p < 0.0001$, $t = 24.42$. Error bars, s.d. ($n = 3$); a.u., arbitrary units; *p* value summary: ****$p$ value < 0.0001, 0.0001 < ***$p$ value < 0.001, 0.001 < **$p$ value < 0.01, 0.01 < *$p$ value < 0.05, $p$ value ≥ 0.05: n.s. Source data are provided as a Source Data file

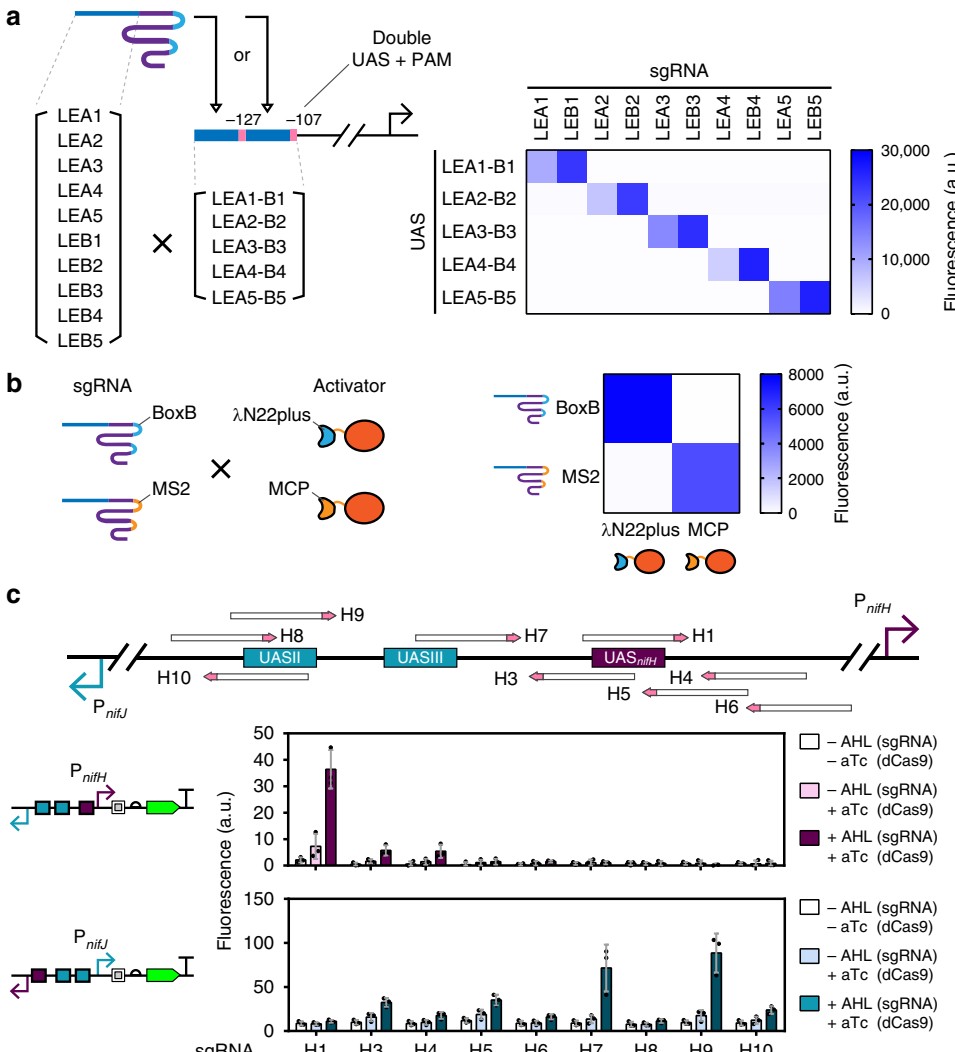

**Fig. 4** Orthogonality tests of the CRISPRa system. **a** Orthogonality owing to UAS-sgRNA specificity. The spacer length for LEA site is 20 bp, and the spacer length for LEB site is 17 bp. Activator expression was driven by the constitutive promoter BBa_J23106. Inducer concentrations used: 1.6 μM AHL (sgRNA), 2.5 ng mL$^{-1}$ aTc (dCas9). **b** Orthogonality between different RNA aptamer-adaptor pairs. Activators expressions were driven by the promoter BBa_J23106. Inducer concentrations used: 0.08 mM arabinose (sgRNA), 2.5 ng mL$^{-1}$ aTc (dCas9). **c** Orthogonal UAS for P$_{nifH}$ and P$_{nifJ}$. Nine selected UAS between the two promoters were tested in activating P$_{nifH}$ and P$_{nifJ}$, respectively (bottom left). Inducer concentrations used: 1.6 μM AHL (sgRNA), 2.5 ng mL$^{-1}$ aTc (dCas9), 3.2 mM rhamnose (activator). Error bars, s.d. ($n = 3$); a.u., arbitrary units. Source data are provided as a Source Data file

The locations of UAS can provide an additional layer of orthogonality. $P_{nifJ}$ and $P_{nifH}$ from the *Klebsiella* genera are divergently transcribed and share a common intergenic region with three UAS for their activator NifA. UAS2 and UAS3 belong to $P_{nifJ}$, and $UAS_{nifH}$ belongs to $P_{nifH}$[41]. We thus hypothesized that our CRISPRa complex would find new and similarly orthogonal UAS. Nine PAMs that spanned across the intergenic region of $P_{nifJ}$ and $P_{nifH}$ were tested in *E. coli* (Fig. 4c). Spacer H1 strongly activated $P_{nifH}$ but not $P_{nifJ}$, and spacer H7 and H9 activated $P_{nifJ}$ but not $P_{nifH}$. The location of these spacers matched well with the locations of their corresponding native UAS and proved that our eukaryote-like CRISPRa system could benefit from orthogonality owing to distinct UAS locations.

**PAM expansion and dynamic range optimization by dxCas9.** Although our CRISPRa system allowed us to target wild-type $\sigma^{54}$-dependent promoters, it demanded the presence of an NGG PAM next to the UAS, which might not be always satisfied. A recent work reported xCas9 as a mutated version of spdCas9 with flexibilities on PAM choices[42]. We thus mutated our dCas9 to dxCas9 based on xCas93.7 and asked if it would allow our CRISPRa to target UAS with non-canonical PAM.

The dCas9 or dxCas9-mediated CRISPRa could activate $P_{pspA}$ with the canonical NGG PAM to similar degrees. Contrary to reported, however, in two cases the dxCas9 could not activate promoters with NNG PAM (Supplementary Fig. 12a, b). We then moved on to test NGT PAM, and CRISPRa using dxCas9 gave remarkable activation (Fig. 5a, Supplementary Fig. 12c). Our goal was to check if dxCas9 could expand available PAM sites on wild-type $\sigma^{54}$-dependent promoters. Therefore, we treated two TGT sequences on the wild-type $P_{pspA}$ as PAM, and attempted CRISPRa on those two UAS using dCas9 or dxCas9. The result confirmed that our CRISPRa using TGT PAM and dxCas9 was functional (Fig. 5b), and hence CRISPRa with dxCas9 allowed more flexible targeting of UAS on different wild-type $\sigma^{54}$-dependent promoters.

In the above experiments, surprisingly, CRISPRa through dxCas9 showed much lower sensitivities to sgRNA leakiness. Their sgRNA-dependent dynamic ranges were improved by an order of magnitude (Fig. 5c). We wondered if this was related to our engineered sgRNA, which contained BoxB aptamers and thus set up a CRISPRi experiment using the dxCas9. We observed a reduction in repression efficiencies mirroring that of activation (Supplementary Fig. 13a–c). It is noteworthy that dxCas9 also worked poorly with sgRNA that contained two mismatches in the scaffold (Supplementary Fig. 14). This suggested that dxCas9 has lower affinities towards its sgRNA or the target DNA and the change in performance probably did not concern our engineered sgRNA scaffold.

Serendipitously, the lowered activation background by dxCas9-based CRISPRa enabled complex circuit designs, which were not attainable by its dCas9 counterpart. This was showcased in a cascaded regulation. dxCas9 could be used to build a two-layer circuit, but when dCas9 was used, background sgRNA expression was sufficiently amplified by the second layer to yield a constant ON state (Fig. 5d). Furthermore, a positive feedback loop, which is self-amplifying in nature and has little tolerance toward input leakiness, could be built from CRISPRa through dxCas9.

This positive feedback loop was derived from the single-layer circuit, but with an additional activator generator driven by the same $\sigma^{54}$ promoter as that of the reporter. Therefore, sgRNA induction should activate not only the reporter but also the extra activator, which in turn would enhance the CRISPRa system per se. As a result, the positive feedback loop using dxCas9 rendered the activation response more digital-like and increased

sgRNA-dependent dynamic range from ~ 50 to 200-fold (Fig. 5e). The same magnitude of fold change was not achieved when dCas9 was used. dxCas9 could suppress the undesired basal activation due to sgRNA leakiness and enhance the dynamic range of the positive feedback loop (Supplementary Fig. 13d).

**CRISPRa for multi-gene expression profile screening.** CRISPR activation and repression is often considered as a useful tool to control metabolic pathway. Many have demonstrated their work through tuning production of violacein, a purple pigment with antitrypanosomal and antitumoral activities[4,43,44]. Yet, most focused on reproducing known effects of flux redirection in this pathway to give different terminal products, by tweaking expression levels of the five enzymes involved. In an industrial setting, pathways of interest are often inadequately characterized with little information for flux optimization. By designing a CRISPRa-based optimization strategy, we could exhaust a metabolic design space, instead of testing flux redirection by trial-and-error.

To demonstrate this, genes encoding the five enzymes (*vioA*—*vioE*) were individually placed downstream of our synthetic $P_{pspA}$ promoters with orthogonal UAS. A total of five corresponding sgRNAs were need for CRISPRa, and three out of the five sgRNAs were combinatorially connected to constitutive promoters of three different strengths, with the remaining two connected to a strong constitutive promoter. The sgRNA generators were then mixed and assembled together in a one-step Golden Gate reaction to yield a sgRNA generator library. The library was thus a collection of diverse transcription patterns of sgRNAs, which would project different sets of activation patterns on the five genes (Fig. 6a). The sgRNA generator library with other CRISPRa components and the $P_{pspA}$ driven violacein-producing genes were then co-transformed into *E. coli*.

We observed that our transformants from the library had colonies with different intensities of purple color, which directly reflected the individual differences in violacein production (Fig. 6b). We then classified colonies into three bins of violacein production, and for each bin, retrieved their individual sgRNA transcription profiles by sequencing the sgRNA generator cassettes from the five colonies (Fig. 6b). We ended up analyzing four profiles for each bin (see Online Methods). Among the three genes experiencing different activation patterns, the promoter strengths for sgRNA activating *vioA* correlated most with violacein production (Fig. 6d), suggesting its role as the limiting factor in the production pathway. It should be noted that the overall strength of the violacein pathway was heavily influence by the level of the dxCas9 and the activator (Supplementary Fig. 15). These two factors thus served as additional gain tuning knobs for optimization of the entire pathway.

Next, a new library was created by mixing sgRNA generators for CRISPRa or CRISPRi, and co-transformed to project more diverse profiles onto the pathway during the profile-scanning process. This addition of CRISPRi introduced possibilities of suppressing the basal expressions of the pathway, and so generated larger differences in violacein production under the same culture condition (Fig. 6c, Supplementary Fig. 16). The ability for CRISPRa and CRISPRi to co-regulate is an advantage of engineering activator onto sgRNA scaffold instead of dxCas9. In addition, the strength of CRISPRa can be tuned independently without affecting that of CRISPRi, by modulating the activator level like tuning the gain knob. This experiment showed potential for intermediate metabolites and pathway redirection scanning in future applications. It is worth noting that, our sgRNA generators already encode all designed transcriptional activation profiles, and therefore it is reusable and optimization of a new pathway would

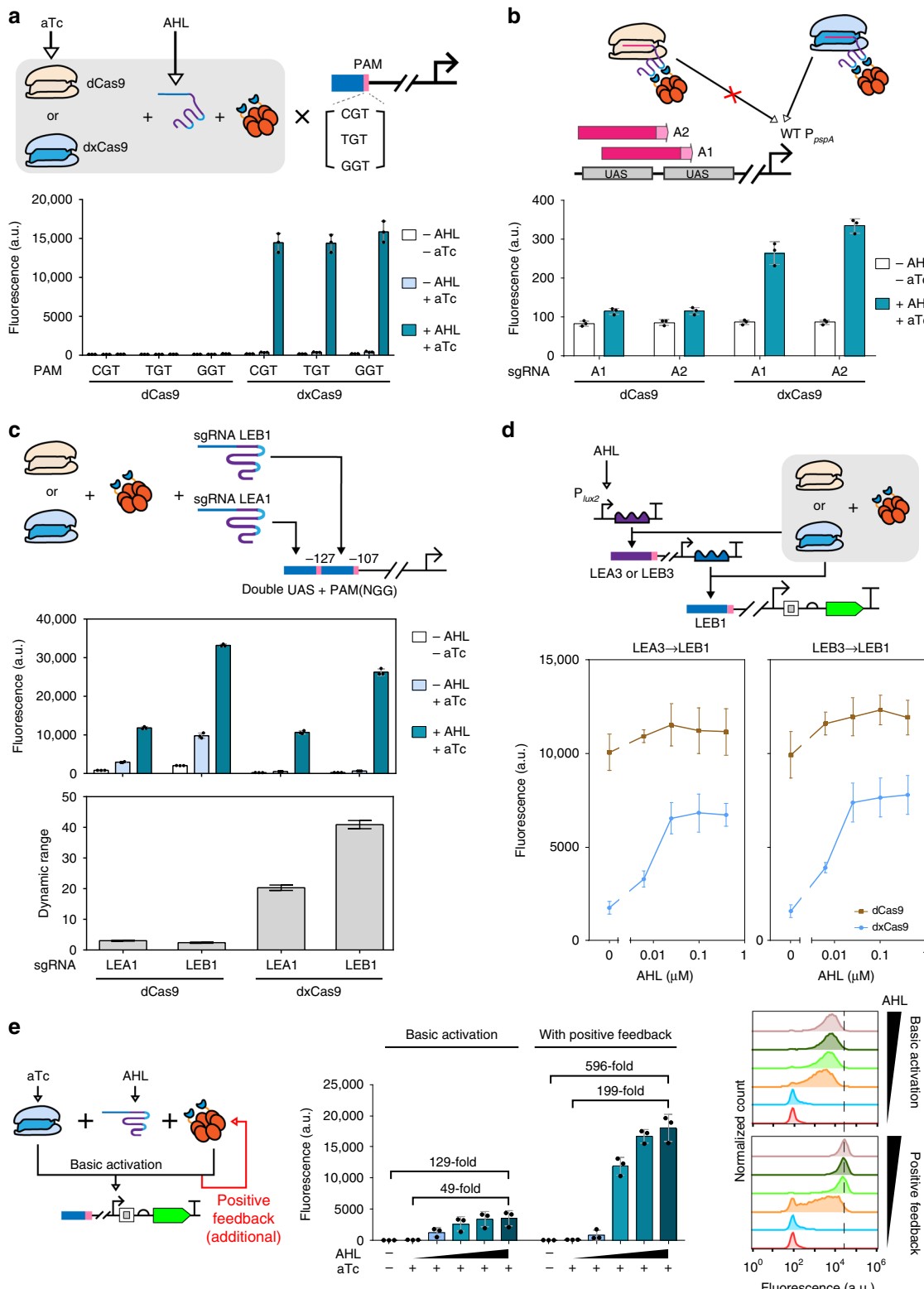

not involve any new library construction—the only step necessary would be the connection of different genes to P_{pspA} with different UAS.

To further demonstrate the application value of this system, we design a "rainbow" device to test if the same multi-gRNA generator library is reusable on a different multi-gene circuit. The library with mixed sgRNA generators for CRISPRa or CRISPRi above was co-transformed with a circuit including three

fluorescent protein genes (the rainbow circuit). The fluorescent protein genes were controlled by the same promoters used to drive *vioA*, *vioD*, *vioC* in Fig. 6. Therefore, each gene may be activated to different extents: (1) No activation (the sgRNA designed for CRISPRi on violacein genes was effectively a mismatch sgRNA), (2) moderate activation, and (3) strong activation (Fig. 7a). As expected, strains isolated from the resulting co-transformed library had a variety of fluorescent

**Fig. 5** CRISPRa using dxCas9 and non-canonical PAM and its lowered background expression. **a** Three different NGT PAM were tested by CRISPRa with dCas9 or dxCas9. The mutated PAM was introduced into the $P_{pspA}$-LEA3B3, which was targeted by sgRNA-LEA3 in this experiment. Activator expression was driven by the constitutive promoter BBa_J23106. Inducer concentrations used: 1.6 µM AHL (sgRNA), 1.25 ng mL$^{-1}$ aTc (dCas9/dxCas9). **b** Two spacers with TGT PAM (A1, A2) within the enhancer region of wild-type $P_{pspA}$ were selected as CRISPRa target UAS. Inducer concentrations used: 1.6 µM AHL (sgRNA), 2.5 ng mL$^{-1}$ aTc (dCas9/dxCas9), 0.4 mM rhamnose (activator). **c** The dynamic range optimization effects of dxCas9 on a synthetic promoter $P_{pspA}$-LEA1B1. Activator expression was driven by the constitutive promoter BBa_J23106. Inducer concentrations: 1.6 µM AHL (sgRNA), 1.25 ng mL$^{-1}$ aTc (dCas9/dxCas9). These data are also shown in Supplementary Fig. 12a. **d** Two-layer CRISPRa using dCas9 or dxCas9. $P_{pspA}$-LEA3B3 was used as the second promoter for the transcription of the sgRNA-LEB1, which activates the third promoter $P_{pspA}$-LEA1B1. **e** A positive feedback loop was constructed using dxCas9 and was compared to a basic activation without feedback. Inducer concentrations used for **d** and **e**: 0, 0.006, 0.025, 0.100, 0.400 µM AHL (sgRNA), 1.25 ng mL$^{-1}$ aTc (dCas9/dxCas9), 0.2 mM rhamnose (activator). The data of basic activation by CRISPRa is also shown in Supplementary Fig. 14. Error bars, s.d. ($n = 3$); a.u., arbitrary units. Source data are provided as a Source Data file

protein expression profiles, which would equate a rainbow color on a population level if the strains were pooled. Sequencing results from the strains revealed good correspondences between fluorescent protein expression profiles and the sgRNA generator units from the multi-gRNA generator (Fig. 7c).

One advantage of our trans-regulated multi-gene control system is its ability to scale the global expression level of all target genes proportionally by tuning the activator induction level —under different CRISPRa efficiencies, absolute expression levels of all fluorescent proteins were altered but the relative expression profiles remained unchanged (Fig. 7b, c).

We also investigated our system's durability, which is important for its application in an industry setting. We serially transferred the culture every 6 hr by diluting the grown culture into fresh medium and measured fluorescence after 6 hr of growth. The result showed that all unique rainbow color profiles from individual strains were stable across the four transfers (Fig. 7d). In addition, there was no detectable metabolic burdens conferred by our CRISPRa-based expression profile scanning device, as evinced by the $A_{600}$ data of this experiment (Supplementary Fig. 17).

## Discussion
In this work, we designed and constructed a CRISPR activation system based on a eukaryote-like enhancer in bacteria. Owing to the unique "unlock" mechanism of AAA + ATPase transcriptional activation domain on $\sigma^{54}$, these promoters inherently possess large dynamic ranges that CRISPRa could harness. Therefore, the CRISPRa output dynamic range was over 1000-fold (dCas9/activator/sgRNA tuning), and was over 70-fold for targeting sgRNA versus non-target sgRNA, which exceeded previous prokaryotic CRISPRa devices. When employed for multi-channel gene regulation, our CRISPRa system utilizing dxCas9 achieved ~ 50-fold output increase when only sgRNA was varied (Fig. 5e), and permitted cascade of genes where no other CRISPRa to date has demonstrated.

$\sigma^{54}$-dependent promoters play important roles in a variety of high-value physiological functions in bacteria, such as nitrogen assimilation and fixation[41,45], pathogenicity[29,46], host colonization[47], motility[46,47], biofilm formation[47], environmental bioremediation[48], and stress responses[45]. In principle, our CRISPRa system can activate any $\sigma^{54}$-dependent promoters. Four typical, high value wild-type $\sigma^{54}$-dependent promoters ($P_{pspA}/P_{hrpL}/P_{nifH}/P_{nifJ}$) from three species have been successfully activated in E. coli. $P_{nifH}$ and $P_{nifJ}$ are two key promoters controlling nitrogen fixation in K. oxytoca and usually require anoxic conditions to be activated[31]. Our CRISPRa system in K. oxytoca could activate these two promoters under aerobic condition, which proved that our device can bypass host regulation mechanisms on natural $\sigma^{54}$-dependent regulation system in non-model bacteria. Such context-independent regulation control provides a powerful

means to study $\sigma^{54}$-mediated gene regulatory networks in situ, which has been difficult to achieve by traditional molecular biology methods.

Similar to activation mechanisms in eukaryotes, our CRISPRa involves long distance regulation and a flexible DNA loop structure. Therefore, unlike previous CRISPRa systems based on the $\sigma^{70}$ factor, this CRISPRa tolerates a wide range of UAS locations (at least 40 bp). It not only accommodates two orthogonal synthetic UAS for input integration on a single promoter, but also provides large flexibility for PAM selection on wild-type $\sigma^{54}$-dependent promoters. Introduction of dxCas9 into our system further enhanced this flexibility by permitting use of non-canonical PAM.

A CRISPRa-based metabolic pathway screening tool has been built to optimize a five-gene pathway of the antitumor chemical violacein in a single step. The scanning and sequencing results demonstrated that the outcomes of violacein production could be explained by transcriptional activation patterns projected onto the pathway. Most importantly, our tool is reusable—the same sgRNA generator library could assay different metabolic pathways if their enzymes are driven by our $P_{pspA}$-derived promoters. A "rainbow" device base on above design was achieved, showing the excellent stability and durability of the multi-gene expression profile control. This showed that our design stores multi-gene expression profile diversities on a universally applicable library, which can be mass produced conveniently. Our tool should be of high value to commercial and industrial applications.

There may be some limitations of this system. Our engineered activator PspFΔHTH::λN22plus was compatible with the $\sigma^{54}$-depedent promoters in both E. coli and K. oxytoca but this compatibility might be challenged in distantly-related microorganisms. In such scenarios, it might be necessary to introduce the heterologous $\sigma^{54}$ factor from E. coli or K. oxytoca, or test other $\sigma^{54}$ activators we developed in this study (engineered PspF, NorR, WtsA). In addition, when activating a wild-type $\sigma^{54}$ promoter, the available PAM sites may limit the maximum activation efficiency of CRISPRa. Nevertheless, whether the activation efficiency is sufficient depends on the specific engineering objectives, which likely vary from one application to another.

Our work also revealed several directions for further study: The fact that spacer H9 for $P_{nifJ}$ was functional in E. coli (Fig. 3a) but less functional in K. oxytoca (Fig. 3b) hinted at fundamental differences in promoter activation processes in the two species. They may involve differences in DNA bending angles, or other unknown factors, which warranted additional studies. In addition, WtsA from E. stewartii, being an orthogonal activator for CRISPRa in E. coli, might give higher activation efficiencies if engineered by directed evolution or other rational design. For dynamic range optimization, saturation mutagenesis on the core sequences of $\sigma^{54}$-dependent promoters may improve their dynamic ranges and output levels. We noted that the expression ratio between PspF ΔHTH::λN22plus and dCas9 may affect the

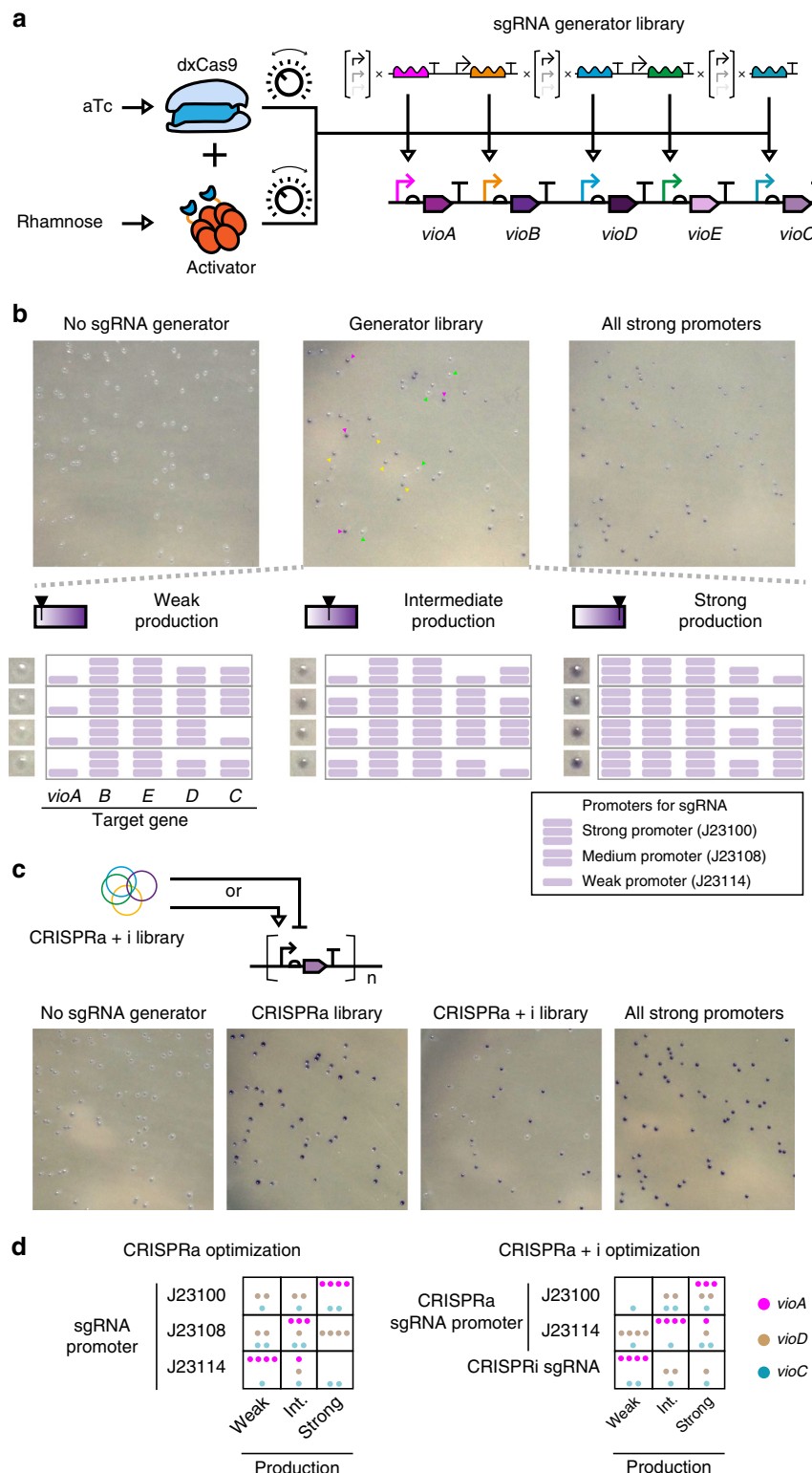

formation of activator hexamer through competition, and this ratio may be further optimized in the future. These further optimizations should realize the full potential of this tool for industrial biotechnology and synthetic biology research.

## Methods

**Strains and growth conditions**. Experiments on activator engineering, heterologous activator scanning, and functional tests in different hosts. The experiments in Fig. 3d, Supplementary Figs. 1f and 2 were performed in the *E. coli* strain TOP10

as indicated. The experiments in Supplementary Fig. 1e were performed in the *E. coli* strain JW3169-1Δ*rpoN730::kan* from the Keio collection[49]. Experiments for P*nif* and P*nifH* activation in *K. oxytoca* was performed in the *K. oxytoca* strain M5a1. All other assays were performed in *E. coli* MC1061Δ*pspF*. *E. coli* MC1061Δ*pspF* was generated via P1 phage transduction using BW25113Δ*pspF739:: kan* from the Keio collection as the donor strain[49].

For growth of *E. coli*, cells were cultured in Lennox's Lysogeny Broth (LB-Lennox) medium (10 g L$^{-1}$ peptone (EMD Millipore), 5 g L$^{-1}$ yeast extract (EMD Millipore), 5 g L$^{-1}$ NaCl (Fisher Scientific)) with appropriate antibiotics. Antibiotics were applied to liquid cultures for assays at final concentrations of: 50 μg mL$^{-1}$ ampicillin (Sigma-Aldrich), 25 μg mL$^{-1}$ kanamycin (Sigma-Aldrich),

**Fig. 6** An expression profiles scanning tool for metabolic pathway. **a** Three genes from the violacein production pathway (*vioA/vioD/vioC*) were chosen as target genes for expression tuning through CRISPRa. Each sgRNA that targeted its cognate promoter could be driven by one of the three constitutive promoters with different strengths and the resulting sgRNA generators were mixed to give a multi-sgRNA generator library. **b** The refactored violacein pathway was co-transformed with the library and cultured on agar plates for 16 hr with 1 μM rhamnose for activator induction and 0.63 ng mL⁻¹ aTc for dxCas9 induction (top middle). For negative control (top left), an empty vector was used in place of the sgRNA generator library. For positive control (top right), the strong constitutive promoter BBa_J23100[27] was used to drive expression of all sgRNA. Colonies with visibly different color intensities were marked by triangles according to their levels of purple color: dark purple (magenta triangles), weak purple (yellow triangles) and white (green triangles). Sequencing results revealed the transcription patterns of sgRNAs in each colony. The number of purple bricks correlates with strengths of the constitutive promoters. Images of the same plates are showed in Supplementary Fig. 15. **c** Pathway optimization by combining CRISPRa and CRISPRi. A new library was constructed by mixing CRISPRi sgRNA (they targeted the coding regions of the three genes and have no RNA aptamers) under a strong promoter (BBa_J23100) and CRISPRa sgRNA under BBa_J23100 or a weak promoter (BBa_J23114). The library was used for profile scanning and results were obtained after 20 hr of culture on agar plates. The images of the same plates are showed in Supplementary Fig. 16. **d** sgRNA transcription profiles were pooled and analyzed. For each regulated gene, the type of sgRNA and their promoter strengths were plotted against the categorized bins of violacein production. Each dot represents a connection that maps the violacein production strength to the sgRNA promoter strength for a given target gene (*vioA/vioD/vioC*). Source data are provided as a Source Data file

and 12.5 μg mL⁻¹ chloramphenicol (Sigma-Aldrich). For circuit construction and selection on agar plates, antibiotics were applied at final concentrations of: 100 μg mL⁻¹ ampicillin, 50 μg mL⁻¹ kanamycin, and 25 μg mL⁻¹ chloramphenicol. For experiments on metabolic pathway profile scanning (Fig. 6, Supplementary Figs. 15 and 16), the agar plates contained 50 μg mL⁻¹ ampicillin and 25 μg mL⁻¹ kanamycin.

For circuit characterization in *E. coli*, a single colony of co-transformant carrying test circuits was picked from an agar plate and suspended in 30 μL of LB medium with appropriate antibiotics. In total, 5 μL of the cell suspension was then inoculated into 195 μL of liquid LB medium with antibiotics in a transparent and flat-bottom 96-well plate (CytoOne). To avoid abnormal growth at the edges of a plate, outermost wells were not used (filled by 200 μL of medium). The plates were cultured at 37 °C, 1000 rpm on a plate shaker (MB100-4A) for 18 hr overnight. The next day, 2 μL of overnight cultures were added into 198 μL of LB medium with appropriate antibiotics and inducers (N-(3-Oxohexanoyl)-L-homoserine lactone, AHL (Sigma-Aldrich), aTc (CAYMAN Chemical), rhamnose (Alfa Aesar), arabinose (Acros Organics)) on a flat and clear bottom 96-well plate with black walls (Greiner Bio-one). The dilution factor was 100-fold. The cultures were then grown for 6 hr at 37 °C, 1000 rpm and then fixed for flow cytometry analysis. All inducers for liquid culture in this research were prepared as 40× stock solutions. Inducers were diluted to 1× final concentrations by adding 5 μL of stock solutions into a final volume of 200 μL of assay volume. There was an exception in metabolic pathway profile scanning (Fig. 6): rhamnose was prepared as a 200× solution and 200 μL of the stock solution was added to 40 mL of molten LB agar to reach 1× final concentration. The aTc was prepared as a 160× solution and 250 μL of the stock solution was added to 40 mL of molten LB agar to reach 1× final concentration. All experiments were done in triplicates. Three colonies were picked and cultured separately on different 96-well plates for induction assays.

*K. oxytoca* M5a1 carries natural ampicillin resistance. 100 μg mL⁻¹ ampicillin was always added to the agar plate during the co-transformation step, and 50 μg mL⁻¹ ampicillin was used for overnight culture. For plasmid maintenance, 50 μg mL⁻¹ kanamycin and 25 μg mL⁻¹ chloramphenicol were used during the final culturing step (6.5 hr) before fixation (without ampicillin).

All reagents (i.e., antibiotics and inducers) were purchased in powder and were prepared by dissolving in double distilled water or ethanol followed by filtration through 0.22 μM syringe filters (Millipore).

**Plasmid circuit construction.** Standard molecular biology protocols were used in this study for circuit constructions. All plasmids and their maps and sequences have been listed in Supplementary Data 1. The typical composition of the plasmids used in this study is illustrated in Supplementary Fig. 18. Key primers used in this study are listed in Supplementary Data 1. Different dCas9 generators, activator generators and reporter circuits were combined on the BioBrick vector pSB4A3[50] for assays in *E. coli*. All sgRNA generators were hosted on a separate and compatible vector p15AC. Variants of each component were chosen according to the specific needs and details of a particular experiment. Qiaspin Miniprep Kit (Qiagen) and Wizard SV Gel and PCR Clean-Up System (Promega) were used in this study for DNA purification.

The dCas9 generator circuit (Supplementary Data 1) was constructed from pdCas9-bacteria (pdCas9-bacteria was a gift from Stanley Qi, Addgene plasmid # 44249[1]). Two synonymous mutations at R63 (CGT > CGC) and R447 (CGA > CGT) were introduced into the dCas9 coding DNA sequence (CDS). One synonymous mutation at S2 (TCT > TCA) was introduced into the TetR CDS, and a base substitution (+29 site of P*tetA*, C > G) was introduced between the P*tetA* transcription start site and the ribosome-binding site (RBS) for *dcas9*. The terminator for the *dcas9* operon was changed to a synthetic transcriptional terminator L3S3P00, and another synthetic terminator L3S3P22 was used for termination of *tetR*. The BioBrick RFC10 prefix and suffix flanked the *dcas9* and *tetR* operons so the entire cluster conformed to BioBrick standards. The dxCas9

generator has the same structure with the dxCas9 CDS mutated by overlap PCR (ProFlex PCR system) (primers provided in Supplementary Data 1). Non-synonymous mutations encoding E1219V, M694I, E543D, E480K, S409I, R324L, and A262T from xCas9 version 3.7[42] were introduced into the dCas9 CDS.

The CDS of PspFΔHTH (1-296 of PspF) was amplified from the genome of *E. coli* TOP10. Through PCR cloning, a BioBrick RBS B0032 was added upstream and the CDS of λN22plus peptide (MNARTRRRERRAEKQAQWKAAN), along with a synthetic terminator L3S3P22 were added downstream. The resulting unit was assembled with either a P*rhaB* promoter or an Anderson promoter through BioBrick RFC10 assembly[51]. Structures and sequences of the Anderson promoters and the standardized P*rhaB* are available in Supplementary Data 1. The three versions of NorR CDS were amplified from the genome of *E. coli* TOP10 (primers provided in Supplementary Data 1). The RBS B0032, the λN22plus CDS and the synthetic terminator L3S3P22 were added by PCR cloning. The NifA2 CDS was amplified from the genome of *K. oxytoca* by PCR (primers provided in Supplementary Data 1), RBS B0032 and terminator L3S3P22 were added by the primer directly. WtsA::GGGS::λN22plus and ZraR::GGGS::λN22plus, both driven by the promoter BBa_J23106[52] and the RBS B0032, terminated by the L3S3P22 terminator, were synthesized by GeneArt with codons optimized for *E. coli*. The engineered HrpRS generator was built from a CRISPRi circuit used in Supplementary Fig. 13. The λN22plus CDS, the RBS B0032, and the terminator L3S3P22 were added or modified by PCR cloning. PspFΔHTH::MCP was constructed by Gibson Assembly, in which the MCP domain was amplified from plasmid pHAGE-EFS-MCP-3XBFPnls which was a gift from Thoru Pederson (Addgene plasmid # 75384[23]). A synonymous mutation N86 (AAT > AAC) was introduced into the CDS of the MCP domain.

The reporter circuits have different promoters. They were all connected to an identical downstream circuit that contains an insulator RiboJ, the RBS B0030, a sfGFP CDS with the SsrA degradation tag (ASV). The wild-type P*pspA* promoter was synthesized by annealing of oligonucleotides (j5 protocol[53]). The synthetic P*pspA* with different UAS were constructed by PCR-based modifications on the UAS regions. The wild-type P*hrpL* was from our previous work[54] and the mutated P*hrpL* with a modified DNA loop, tested in the PspFΔHTH::dCas9 and the CRISPRi assay, was constructed by reverse PCR. Promoters P*nifH* and P*nifJ* were amplified from the genome of *K. oxytoca* M5a1 by PCR (primers in Supplementary Data 1). All buffer terminators and hybrid promoters were constructed by annealing oligonucleotides or PCR.

The CRISPRi circuit includes a dCas9 generator, a HrpRS generator (RA32T in Supplementary Data 1) and a reporter circuit. RBS B0030 and B0032 drive the translation of *hrpR* and *hrpS*, respectively, in a bicistronic operon that is terminated by the terminator B0015. A mutated P*hrpL* was used in the reporter circuit. Its core region was mutated (−2, T > A; −5, G > A; −14, C > T; −21, T > C) to improve its maximum output. Furthermore, the −65 & −66 sites (AA > CC) and the −90 & −91 sites (TA > CC) of this promoter were mutated to create NGG PAM within the DNA loop region. The 16 bp fragment between the −24 box and the IHF region (−42 to −63), and the 19 bp fragment from −70 to −88 were replaced by random sequences (Supplementary Data 1) for orthogonal CRISPRi targeting.

For constructions of sgRNA generators, the vector p15AC was amplified from pdCas9-bacteria and was flanked by two *BbsI* sites, which would create sticky ends for *EcoRI* and *SpeI*, respectively. The sgRNA scaffolds with BoxB or MS2 aptamers were synthesized by annealing oligonucleotides. Each generator contains a P*lux2* or P*BAD* promoter and a L3S2P21 terminator. Whenever necessary, the spacer sequence was modified by PCR cloning. All buffer terminators were inserted 5' of sgRNA spacer by PCR cloning. For modularity test of buffer terminators, the B0062-R terminator was synthesized by annealing oligonucleotides and assembled with the insulator RiboJ, which was then inserted into sgRNA generator. Modifications of the sgRNA scaffold, including introduction of mismatches, were performed by PCR cloning.

For heterologous expression of CRISPRa in *K. oxytoca* M5a1, the activator generator was driven by the promoter BBa_J23106, as P*rhaB* is non-functional in

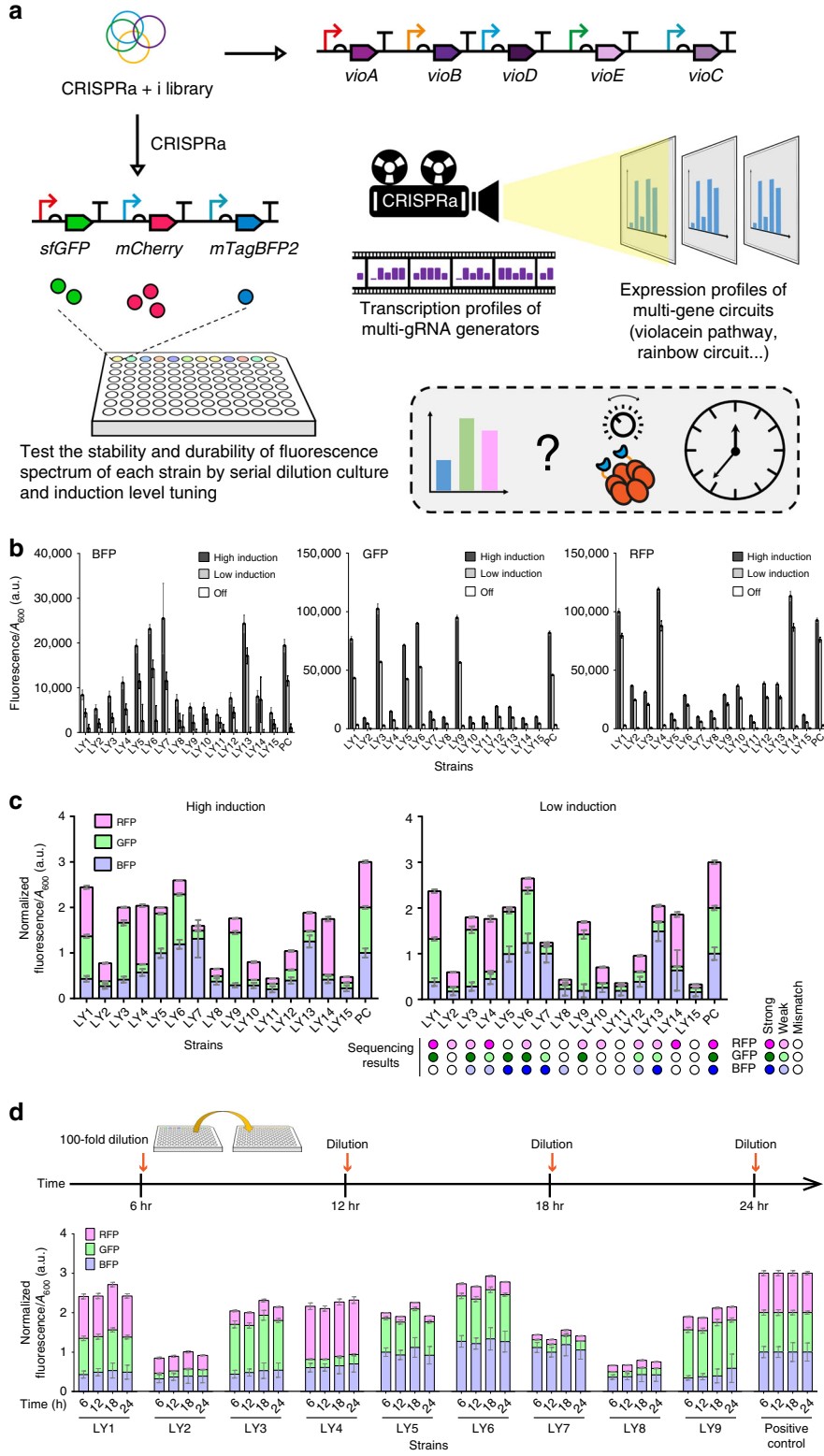

K. oxytoca. Circuits on pSB4A3 in E. coli were migrated into pSB4K5[55] owing to the endogenous ampicillin resistance of K. oxytoca M5a1. No vector change was necessary for sgRNA generators.

All constructs in this study were sequenced by Sanger sequencing service provided by Source Bioscience or MRC PPU DNA sequencing and services.

**Metabolic pathway profile scanning.** Libraries of tandem sgRNA generators for transcriptional activation profile scanning were constructed through the following

steps: First, basic sgRNA generator units with three components were built, each includes an upstream terminator, an Anderson promoter and the sgRNA sequence. The upstream terminator, with BbsI restriction sites that would create unique overhangs (Supplementary Data 1), and Anderson promoters with different strength were made by annealing synthesized oligonucleotides. They were then ligated upstream to an amplified and digested sgRNA sequence on the vector pSB1K3[56] (sgRNA fragments were digested by XbaI and PstI). Second, sgRNA generator units that contain an identical spacer but different promoters (BBa_J23100 /BBa_J23108 /BBa_J23114)[27] were mixed in equimolar ratios to give one set of sgRNA generator

**Fig. 7** The reusability of the multi-gRNA generator library, and the stability of expression profiles. **a** The multi-gRNA generator library (with sgRNA for CRISPRa and CRISPRi) we used for violacein pathway could be used on other target circuits. The CRISPRa device can project the transcription profiles of each multi-gRNA generators to target circuits and generate a variety of gene expression pattern. Three different fluorescent protein genes were used as reporters in this experiment. The genes *sfGFP*, *mCherry* and *mTagBFP2* were controlled by the same artificial σ[54] promoters for *vioA*, *vioD*, *vioC* respectively. Each gene may be activated by CRISPRa to different extents: (1) No activation (the sgRNA designed for CRISPRi on violacein genes was effectively a mismatch sgRNA), (2) moderate activation, and (3) strong activation. **b** Absolute fluorescence/$A_{600}$ values of the three reporters detected from each strain. Left: values from BFP channel; Middle: GFP; Right, RFP. For high-induction condition, 1.25 ng mL$^{-1}$ aTc (dxCas9), 0.4 mM rhamnose (activator) were used. For low induction condition, 1.25 ng mL$^{-1}$ aTc, 0.1 mM rhamnose were used. 1.25 ng mL$^{-1}$ aTc, 0 mM rhamnose were used for OFF state. The PC strain was a positive control strain that carried a multi-gRNA generator with strong promoter BBa_J23100 for all sgRNA for CRISPRa. **c** PC strain-normalized fluorescence/$A_{600}$ values, which shows the relative proportions of the three fluorescent proteins in each strain under different induction conditions. Legend for colored circles beneath the data bars: dark color, the sgRNA transcription is driven by strong promoter; light color, driven by weak promoter; white color, mismatch sgRNA. Information for colored circles come from the sequencing results of each strain. **d** Serial transfers (100-fold every 6 hr) and induced growth up to 24 hr. 1.25 ng mL$^{-1}$ aTc (dxCas9), 0.4 mM rhamnose (activator) were used in this test. The bar chart shows PC strain-normalized fluorescence/$A_{600}$ values. Error bars, s.d. ($n = 3$); a.u., arbitrary units. Source data are provided as a Source Data file

units. Each set corresponds to a single target gene in which expression would be tuned. Five sets of sgRNA generator units were then mixed and assembled in a one-step Golden Gate Assembly system into pSB4A3. It should be noted in Fig. 6, two out of the five sets had no variation in promoter choice and only used the promoter BBa_J23100. Third, the entire Golden Gate Assembly reaction was transformed into *E. coli* TOP10 and cultured on an agar plate with ampicillin for over 20 hr. All colonies on plate (with > 250 colonies, which should cover 27 possible combinations) were scraped and washed down from the plate and subjected to plasmid extraction, which yielded a crude library. Fourth, the crude library was cleaned up by size selection—plasmids were digested by *Eco*RI and *Pst*I, and then resolved on an agarose gel by gel electrophoresis. Digested fragments, with sizes that correspond to the desired assembly product, were excised and self-ligated, which was then transformed into *E. coli* TOP10 again. The final library was obtained by scrapping all transformants followed by plasmid extraction. The positive control sgRNA generator circuit was made by following the same steps with the five kinds of sgRNA generator units that only used the promoter BBa_J23100.

The above steps were repeated in generating the CRISPRa + i profile scanning library with the following exception: When building a sgRNA generator unit set, sgRNA generators that contain standard sgRNA scaffold plus spacers that target CDS of *vioA/vioD/vioC* were added to the mixture of CRISPRa sgRNA generators.

To construct the violacein production pathway for optimization on metabolic engineering, the CDS of *vioA*, *vioB*, *vioE*, *vioD* were amplified from pΔ_lvl2_vioABDE[57] and the CDS of *vioC* gene was amplified from pKMV-VioC (Addgene plasmid #65349[58]). The RBS B0030 and five different terminators were added to each gene through PCR cloning (L3S2P21 for *vioA*, L3S2P11 for *vioB*, L3S3P22 for *vioE*, L3S3P21 for *vioD*, and L3S2P55 for *vioC*). These translational units for VioA/B/D/E/C were placed downstream of P$_{pspA}$-LEA1B1, P$_{pspA}$-LEA2B2, P$_{pspA}$-LEA3B3, P$_{pspA}$-LEA4B4, and P$_{pspA}$-LEA5B5, respectively. They were then assembled with a dxCas9 generator and the PspFΔHTH::λN22 generator on the vector pSEVA221 (GenBank: JX560327).

Circuits including CRISPRa device, the sgRNA library and the violacein production pathway were co-transformed into *E. coli* MC1061ΔpspF, which were then spread on LB-Lennox agar plates with ampicillin, kanamycin, aTc and rhamnose. Three kinds of plates with different inducer concentrations were made and used for culturing co-transformants: (1) 5 μM rhamnose + 0.63 ng mL$^{-1}$ aTc, (2) 1 μM rhamnose + 0.63 ng mL$^{-1}$ aTc, and (3) 0 μM rhamnose + 0.63 ng mL$^{-1}$ aTc. All plates were incubated at 37 °C overnight. Pictures of the agar plates with colonies were then taken by a digital SONY NEX-5R camera under normal lighting conditions in a laboratory after incubation durations of 14, 16, 18, and 20 hr. For post-acquisition processing, brightness and contrast of photos were adjusted in Photoshop CS3. Each photo was then scaled using a five pence coin that was taken in the same shot so that colony sizes across different plates would be comparable. All scaled photos were cropped to areas of interest. Colonies to be picked and sequenced were selected from an area of interest at 16 (CRISPRa) or 20 (CRISPRa + i) hr post-incubation and their positions were recorded. They were then picked and inoculated after the last photograph time point has passed.

For the expression profile test on the rainbow device, the above CRISPRa + i library was co-transformed with the rainbow circuit with the same steps. The agar plate did not include inducers. Colonies were picked and suspended in 30 μL of LB medium. For each strain, 5 μL of cell suspension was added into 195 μL LB medium with ampicillin and kanamycin for overnight (18 hr) liquid culture in a 96-well plate, which then was diluted 100-fold into fresh medium with antibiotics and inducers and grown under 37 °C, 1000 rpm for 6 hr for data collection. Part of the cultured strains (LY1–LY9) were serially transferred every 6 hr (2 μL of grown culture into 198 μL of fresh medium) with antibiotics and inducers up to 24 hr. The plate was read every 6 hr after each transfer by a plate reader (BMG FLUOstar fluorometry).

**Gene expression assays.** For all the experiments except rainbow device, only one standardized reporter sfGFP was used to characterize promoter outputs in this work. All fluorescence data (except the rainbow device and the experiment of

Supplementary Fig. 1f) was measured in an Attune NxT Flow Cytometer (equipped with Attune NxT Autosampler). For fixation, 2 μL of cultured bacteria was added to 198 μL 1 × phosphate-buffered saline (K813-500ML, VWR) in a 96-well U-bottom plate (Thermo Fisher Scientific) with 1 mg mL$^{-1}$ kanamycin and the samples were stored in 4 °C for at least 1 hr before data acquisition.

Fixed samples were read by the Flow Cytometer, equipped with 488 nm excitation laser and 530/30 nm emission filter for green fluorescence measurements. For each sample reading, the cytometer was instructed to wait for 4 s before recording single cell fluorescence, and at least 10,000 gated events were recorded using a flow rate of 25 μL min$^{-1}$. Flow cytometry data were analyzed by FlowJo (v7.6.1) and appropriate forward and side scatter gates were applied to all samples for each experiment. The geometric means of fluorescence were taken as population fluorescence. For the UAS shift test in Fig. 2a, median values were taken as population fluorescence to avoid impact of extreme values. For rainbow device, we used a plate reader (BMG FLUOstar fluorometry) to collect fluorescence and $A_{600}$ data, and calculate fluorescence/$A_{600}$.

For the measurement of expression profiles of the rainbow strains and the experiment of Supplementary Fig. 1f, data were acquired from a plate reader equipped with a 405–10 nm excitation laser and a 460 nm emission filter for blue fluorescence measurements (Gain 1372), a 485 nm excitation laser and a 520 nm emission filter for green fluorescence measurements (Gain 1263 for the rainbow strains; Gain 700 for the experiment of Supplementary Fig. 1f), and a 584 nm excitation laser and a 620–10 nm emission filter for red fluorescence measurements (Gain 2240).

**Absorbance ($A_{600}$) assays.** Cell densities, measured by absorbance ($A_{600}$), were used as a proxy of cellular burden. $A_{600}$ values were measured using a plate reader (BMG FLUOstar fluorometry). To set up blanks for measurements, 200 μL of liquid LB medium was added to a blank well without inoculation, in which the volumes to be occupied by inducers were filled with water. This step would be done at the same time when cells were diluted to give assay cultures. Therefore, effects of evaporation and inducer dilution should apply equally to both the blanks and the cell cultures. The $A_{600}$ values were processed and analyzed in Microsoft Excel 2013.

**Quantitative RT-PCR.** The strains were cultured in 200 μL Lennox's LB medium with appropriate antibiotics under 37 °C 1000 rpm overnight (16–18 hr). They were then diluted 100-fold into 1 mL fresh medium with appropriate antibiotics and inducers in a 96-well deep-well plate (Starlab). The diluted culture was incubated under 37 °C 1000 rpm for 6 hr. Then the total RNA was extracted by the Aurum™ Total RNA Mini Kit (Bio-Rad). The RQ1 RNase-Free DNase (Promega) was employed to digest the residual genomic DNA, and the iScript™ Reverse Transcription Supermix (Bio-Rad) was used to generate cDNA in 10 μL reaction systems through reverse transcription. The iTaq™ Universal SYBR® Green Supermix was used for qPCR in 10 μL reaction systems with 60 °C as the annealing temperature, and 30 s were set as the extension time. The housekeeping gene *rho* was used as the reference gene. All the genes in RT-qPCR experiments were isolated from the genome of *E. coli* and were cloned on a high copy-number plasmid pSB1K3 as the standard sample for standard curve generation. The qPCR reactions were run on a LightCycler 480 qPCR system (Roche). The relative expression levels of target genes were calculated using the LightCycler 480 software and Excel 2013 by the Pfaffl method (normalized by the reference gene *rho*). The primers for gene isolation and qPCR are provided in Supplementary Data 1.

**Calculation of fluorescence intensities.** The RT-qPCR experiment includes 6 biological repeats. All attempts at replication were successful. All geometric mean of fluorescence of a population were corrected by background fluorescence of cells with suitable empty plasmids grown in identical conditions but without inducers. Corrected fluorescence values were then averaged in GraphPad Prism 7.05 (except rainbow device) to give the medians or means with standard deviations (SD),

which were used for plotting graphs. There were two exceptions: the data for heat maps (Fig. 4a, b) and the rainbow device (Fig. 7): For heat maps, means were calculated in Microsoft Excel 2013 and then imported into GraphPad Prism for plotting the heat maps. Two-tailed $t$ tests were performed using GraphPad Prism 7.05. For the rainbow device and the experiment of Supplementary Fig. 1f, the mean of fluorescence values and OD values were corrected by a blank negative control (with corresponding volumes of medium and with water replacing inducers). The fluorescence/$A_{600}$ for each well was then calculated. The fluorescence/$A_{600}$ value was corrected by that of a fluorescence-free strain. All the calculation was done in Microsoft Excel 2013. Error calculation follows the propagation of error.

**Calculation of cell densities (absorbance $A_{600}$).** All $A_{600}$ values were corrected by blank absorbance of liquid LB medium without cells. Corrected $A_{600}$ values were then averaged in GraphPad Prism 7.05 to give means and standard deviations (SD), which were used for plotting graphs.

**Calculation of dynamic ranges.** For all dynamic range calculations, the means and standard deviations (SD) of geometric mean fluorescence from all samples were first calculated by the built-in formulae AVERAGE and STDEV in Microsoft Excel 2013. The calculated mean was then corrected by the mean of background fluorescence of cells. Error propagation was calculated by the following formula:

$$SD_{cor} = \sqrt{SD(x)^2 + SD(nc)^2} \qquad (1)$$

$SD_{cor}$, standard deviation of corrected mean; $SD(x)$, SD of the three geometric mean fluorescence values from particular sample x; $SD(nc)$, SD of that from negative control sample.

The dynamic range $DR$ was calculated by the corrected mean value of fluorescence:

$$DR = (Fluo(on) - Fluo(off))/Fluo(off) \qquad (2)$$

$Fluo(on)$, the corrected means of three geometric mean fluorescence values from samples under on state; $Fluo(off)$, the corrected means of three geometric mean fluorescence values from samples under off state; $DR$, dynamic range.

The standard deviation of $DR$ ($SD(DR)$) was calculated by the uncertainty propagation formula:

$$SD(on - off) = \sqrt{SD_{cor}(on)^2 + SD_{cor}(off)^2} \qquad (3)$$

$$SD(DR) = \sqrt{(SD(on-off)/(Fluo(on) - Fluo(off)))^2 + (SD_{cor}(off)/Fluo(off))^2} \times DR \quad (4)$$

The final mean ($DR$) and standard deviation ($SD(DR)$) values were imported into GraphPad Prism 7.05 for plotting graphs.

**Sequencing analysis of metabolic activation profile scanning.** For CRISPRa-based metabolic activation profile scanning, and for each bin of purple color intensity, five colonies were identified and marked after 16 hr of growth on agar plates. The same was applied to CRISPRa+i based profile scanning, but colonies were identified and marked after 20 hr of growth. For each scan, those previously marked colonies were picked and each was inoculated into 10 mL of LB medium with 100 μg mL$^{-1}$ ampicillin in a 30 mL universal container (Starlab). After overnight culture at 37 °C, 160 rpm, plasmids were extracted from each culture and were sequenced by Sanger sequencing (Source BioScience). The sequencing results were aligned to known sequences of sgRNA generators by the software SnapGene 3.2.1 to recover the identities of the promoters of the sgRNA generator, or the identity of the CRISPRi sgRNA. Results that failed to give proper reads or indicated mixed clones were excluded from downstream analysis. For fair comparison, each bin used the same number of colonies for analysis.

**Reporting summary.** Further information on research design is available in the Nature Research Reporting Summary linked to this article.

## Data availability

The previously constructed plasmids pdCas9-bacteria (Addgene #44249), pHAGE-EFS-MCP-3XBFPnls (Addgene #75384), and pKMV-VioC (Addgene #65349) used in this study are available from Addgene. Key plasmids constructed in this study are available from Addgene at https://www.addgene.org/Baojun_Wang/. All other data and materials from this work are available from the corresponding author upon reasonable request. The source data underlying Figs. 1c–e, 2a–d, 3a–d, 4a–c, 5a–e, 6b, c, and 7a–d, and Supplementary Figs 1a–f, 2a, b, 3a, b, 4a, b, 5a–f, 6a, b, 7a–c, 8a, b, 9a, b, 10a, b, 11a, b, 12a–c, 13a–d, 14–17 are provided as a Source Data file.

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

## Acknowledgements

We thank Trevor Y. H. Ho for his help in preparing the manuscript and valuable suggestions. We thank Dr Jorg Schumacher (Imperial College London) for supplying the *Klebsiella oxytoca* M5a1 strain, Andreas Andreou (University of Edinburgh) for providing plasmid pΔ_lvl2_vioABDE and Dr Attila Molnar for critical reviewing of the manuscript. This work was supported by the UK Biotechnology and Biological Sciences Research Council grant [BB/N007212/1], UK Research and Innovation [MR/S018875/1], Leverhulme Trust grant [RPG-2015-445], and Wellcome Trust Seed Award in Science [202078/Z/16/Z]. X.W. was supported by scholarships from the China Scholarship Council and the Scottish Universities Life Sciences Alliance.

## Author contributions

B.W. and Y.L. conceived the study. Y.L. and B.W. designed the experiments. Y.L. and X.W. performed the experiments and data analysis. All authors took part in the interpretation of results and preparation of materials for the manuscript. B.W. and Y.L. wrote the manuscript.

## Additional information

**Competing interests:** The authors declare no competing interests.

