## [Peer Review File · Nature Communications]

Editorial Note: This manuscript has been previously reviewed at another journal that is not operating a transparent peer review scheme. This document only contains reviewer comments and rebuttal letters for versions considered at Nature Communications. Mentions of prior referee reports have been redacted.

Reviewers' Comments:

Reviewer #1:

Remarks to the Author:

The authors have answered some of my questions but some important questions that would improve the manuscript remain unanswered. Definitely I think this manuscript should be published in Nature Comms, but answering my questions may help making an outstanding paper.

Major comments:

- Q1: Ok

- Q2, 3, 4: Ok

- Q5: Ok, but mention the comments on co-expressing PspF Δ HTH:: λ N22plus and PspF Δ HTH in the discussion.

- Q6 (comment 1): The authors didn't understand my comment "It is mentioned that the PspA protein could be inhibiting PspF in Top10, but the strain MC1061 Δ pspF also has the pspA intact (and prone to activation) and a similar inhibition would be expected." and their answer is not relevant. Let me rephrase it: "PspA protein controlled by PspA promoter on the genome, which is known to allosterically inhibit PspF". Could the PspA protein inhibit (through protein-protein interactions!) the PspF Δ HTH:: λ N22plus?

- Q6 (comment 2): Consider also the following: The binding between PspF Δ HTH:: λ N22plus and PspF will sequester PspF molecules (again protein-protein interactions), impairing the activation of your engineered promoter in TOP10. These comments will enrich the discussion.

- Q7: This question has not been answered. I disagree with the use of a terminator after the promoter and pretend it will be functional unless it is demonstrated. If you insist, then it is possible to design an experiment for your system where you obliterate the terminator by a suitable mutation in the poly-U sequence, which should have a minimal effect on transcript stability. You asked for a reference, see for instance Telesnitsky AP, Chamberlin MJ. Sequences linked to prokaryotic promoters can affect the efficiency of downstream termination sites. *J Mol Biol.* 1989 Jan 20;205(2):315–30. Note also that sequences immediately downstream the promoter (particularly for those promoters truncated/characterised before the +1) may affect the initiation of transcription. I'm not necessarily asking for more experiments to be done, but for the readers to be presented with a sound strategy. Some readers only interested on the optimisation of the expression of sgRNAs may also be interested in this manuscript.

- Q8: I had meant aTc/rhamnose in my question but you already tested various rhamnose levels in Fig. 1c. Can you provide the dynamic range in those experiments with lower levels of rhamnose? You already have the data. Add a comment on my mismatch remark in the discussion (including the reference), as it could be useful for the reader.

- Q9: Ok

- Q10, Q11: To clarify my comment. If you have a sgRNA targeting the operator site of a transcription factor (TF) then the DNA encoding such sgRNA may also contain the binding site (it will contain the intersection of the 20bs of the guide sequence with the operator). As orientation is not a problem for TF and the guide sequence is located at the 5' side then you may end up with a potential binding site for the TF just downstream the promoter, which very often leads to its downregulation. Therefore, you may have an additional regulation in the sgRNA promoter when in presence of the TF. For instance, the promoter pPspA-mut contains a target guide sequence overlapping the UAS equal to "TTAGTGTAATTCGCTA" (Suppl. Table 3), this sequence is (of course) also found in sgRNA-mut (Suppl. Table 5). gRNA use PAMs to avoid self-targeting but TFs always bind. I don't think this invalidates any results, but it should be acknowledged in the discussion.

This is why I had said in Q11 "Using the native UAS seems bad to me".

- Q12, Q13: Ok

- Q14: I understand that you didn't engineer any cooperativity but Fig. 5d shows a more sigmoid behavior for dxCas9 than for dCas9. This behavior has never been seen with a dCas9 not even with dimerization (cited reference). This is a big deal and because of that I'm asking again why the sgRNA activation shows such non-linear behavior? It contradicts all previous known (to me) results. A discussion of this point would also be useful to the reader.

- Q15: Ok

- Q16: Please add this to the discussion, it will be useful for the readers.

- Q17: Ok

- Please use logarithmic scale in all fluorescence barplots wherever you have bars of small size (indistinguishable from zero, which is not informative at all).

- Minor comments:

- Fig 1: the labelling of subpanels is awkward, I would use c, d, e instead of d, e, c

- Page 4 and other places: replace "chimeric hexamers" by "heterohexamers " throughout the text as chimeric constructions are usually used for protein fusions.

Reviewer #2:

Remarks to the Author:

N/A

Point-by-point responses (in blue) to reviewers' comments (in black):

REVIEWERS' COMMENTS:

Reviewer #1 (Remarks to the Author):

The authors have answered some of my questions but some important questions that would improve the manuscript remain unanswered. Definitely I think this manuscript should be published in Nature Comms, but answering my questions may help making an outstanding paper.

Major comments:

- Q1: Ok

- Q2, 3, 4: Ok

- Q5: Ok, but mention the comments on co-expressing PspF Δ HTH:: λ N22plus and PspF Δ HTH in the discussion.

We are happy to follow as suggested. We have added the following relevant texts into the discussion: 'We noted that the expression ratio between PspF Δ HTH:: λ N22plus and dCas9 may affect the formation of activator hexamer through competition, and this ratio may be further optimized in the future.'

- Q6 (comment 1): The authors didn't understand my comment "It is mentioned that the PspA protein could be inhibiting PspF in TOP10, but the strain MC1061 Δ pspF also has the pspA intact (and prone to activation) and a similar inhibition would be expected." and their answer is not relevant. Let me rephrase it: "PspA protein controlled by PspA promoter on the genome, which is known to allosterically inhibit PspF". Could the PspA protein inhibit (through protein-protein interactions!) the PspF Δ HTH:: λ N22plus?

We thank Reviewer #1 for clarifying the question, and we would like to elaborate our answer below:

PspA could inhibit both wild type PspF or PspF Δ HTH:: λ N22plus. However, the *pspA* gene on the genome of *E. coli* is driven by σ^{54} -promoter P_{pspA} , which only works when the wild type PspF exists. In the strain MC1061 Δ pspF, there is no wild type PspF activator, PspF Δ HTH:: λ N22plus would not activate the *pspA* gene unless we use a particular sgRNA to target the P_{pspA} promoter. So if the sgRNA could not target WT P_{pspA} , the PspA should not be expressed in the strain MC1061 Δ pspF.

The situation in TOP10 and MC1061 Δ pspF is different. In TOP10, there is a closed loop negative feedback regulation on the expression of PspA. We did not investigate whether *pspA* had background expression (without CRISPRa) or whether it was activated by heterohexamers of PspF Δ HTH:: λ N22plus and PspF, but either way its activity should be higher than that in MC1061 Δ pspF. In MC1061 Δ pspF, this negative feedback regulation loop does not exist and the *pspA* gene should theoretically always stays at the 'OFF' state.

While it is possible that drop in activation activity in TOP10 was due to PspA inhibiting both wild type PspF or PspF Δ HTH:: λ N22plus via protein-protein interaction, we are inclined towards the explanation that a complete negative feedback regulation in the stain TOP10 negatively influenced the function of PspF Δ HTH:: λ N22plus and caused the lower expression level (see below schematic).

- Q6 (comment 2): Consider also the following: The binding between PspF Δ HTH:: λ N22plus and PspF will sequester PspF molecules (again protein-protein interactions), impairing the activation of your engineered promoter in TOP10. These comments will enrich the discussion.

We thank Reviewer #1 for pointing this out. However, we have a different anticipation on the consequences brought by the heterohexamer formation of PspF Δ HTH:: λ N22plus and PspF. If the binding between PspF Δ HTH:: λ N22plus and PspF could impair the activation on our engineered promoter, it must be based on a premise that the heterohexamers of PspF Δ HTH:: λ N22plus and PspF is not functional for binding to UAS or unlocking the σ^{54} factor. Otherwise the effect will be the opposite. The result in **Supplementary Figure 1f** has already indicated that the heterohexamers of PspF Δ HTH:: λ N22plus and PspF is functional because they could unlock the σ^{54} factor, and for UAS targeting only two free λ N22 adaptors were needed. We believe it is likely that the presence of WT PspF enhances the activation efficiency (if we don't consider PspA in the cell).

- Q7: This question has not been answered. I disagree with the use of a terminator after the promoter and pretend it will be functional unless it is demonstrated. If you insist, then it is possible to design an experiment for your system where you obliterate the terminator by a suitable mutation in the poly-U sequence, which should have a minimal effect on transcript stability. You asked for a reference, see for instance Telesnitsky AP, Chamberlin MJ. Sequences linked to prokaryotic promoters can affect the efficiency of downstream termination sites. *J Mol Biol.* 1989 Jan 20;205(2):315–30. Note also that sequences immediately downstream the promoter (particularly for those promoters truncated/characterised before the +1) may affect the initiation of transcription. I'm not necessarily asking for more experiments to be done, but for the readers to be presented with a sound strategy. Some readers only interested on the optimisation of the expression of sgRNAs may also be interested in this manuscript.

We thank Reviewer #1 for pointing this out and providing the reference, we agree with the opinion that that sequences immediately downstream the promoter could affect its function. So we modify our manuscript as following: 'we inserted various known terminators as buffer sequences" (named as "buffer terminators") between P_{lux2} and sgRNA to interfere the function of the promoter and the output ranges of sgRNA transcription.'

- Q8: I had meant aTc/rhamnose in my question but you already tested various rhamnose levels in Fig. 1c. Can you provide the dynamic range in those experiments with lower levels of rhamnose? You already have the data. Add a comment on my mismatch remark in the discussion (including the reference), as it could be useful for the reader.

We would like to clarify that we do not have the 'dynamic range' data, defined as sgRNA "ON/OFF" condition for the various rhamnose (activator) levels in Fig 1c. The data in Fig1c only includes data of the sgRNA induction states under a gradient of rhamnose, and do not have the accompanying 'OFF state' data. We will be grateful if Reviewer #1 could understand that, while we agree the proposed experiment would be helpful for fine-tuning the circuit, it is not directly relevant to our message in Fig 1c. Therefore, for the time being we do not intend to repeat that experiment with more controls. There could be many more methods in optimizing CRISPRa but with the number of words limit we are afraid we could not exhaust all options in this paper.

For the same reason, we opt to leave out the strategy of using mismatches in spacer sequences in our discussion, despite our agreement that it is feasible. We did not go in length to review the details on optimization strategies in our discussion, and only recommend dxCas9 because our data suggested that it is the most convenient and generally applicable method.

- Q9: Ok

- Q10, Q11: To clarify my comment. If you have a sgRNA targeting the operator site of a transcription factor (TF) then the DNA encoding such sgRNA may also contain the binding site (it will contain the intersection of the 20bs of the guide sequence with the operator). As orientation is not a problem for TF and the guide sequence is located at the 5' side then you may end up with a potential binding site for the TF just downstream the promoter, which very often leads to its downregulation. Therefore, you may have an additional regulation in the sgRNA promoter when in presence of the TF. For instance, the promoter pPspA-mut contains a target guide sequence overlapping the UAS equal to "TTAGTGTAATTCGCTA" (Suppl. Table 3), this sequence is (of course) also found in sgRNA-mut (Suppl. Table 5). gRNA use PAMs to avoid self-targeting but TFs always bind. I don't think this invalidates any

results, but it should be acknowledged in the discussion. This is why I had said in Q11 "Using the native UAS seems bad to me".

We perceive that there might be a misunderstanding here, on whether the spacer sequence itself is sufficient for recruitment of the activator. It is true that we do not have experimental evidence on whether the wild type PspF could bind to sgRNA generator circuit. However, the spacer sequence itself is considerably much shorter than the minimal UAS sequence determined experimentally in previous research. It has been shown that there are two UAS sites for PspF hexamer binding, and the 'GTgtaattcgctaACT' site should be the UAS I site by DNase I footprint experiment¹. Generally, scientists see the two UASs together as the UAS region for PspF binding, and deleting one of them would significantly decrease the activation efficiency². In our case, the spacer sequence is not long enough to provide the complete UAS I site. Hence we reasoned that it is very unlikely that PspF would bind to the incomplete UAS I sequence. Yet, we agree that we could not rule out this possibility. Therefore, we incorporated this as a possible reason for lower CRISPRa output in the TOP10 strain by adding below sentence in the main text:

'It is worthwhile to mention that the lower output level in TOP10 may also be caused by other unknown mechanisms.'

- Q12, Q13: Ok

- Q14: I understand that you didn't engineer any cooperativity but Fig. 5d shows a more sigmoid behavior for dxCas9 than for dCas9. This behavior has never been seen with a dCas9 not even with dimerization (cited reference). This is a big deal and because of that I'm asking again why the sgRNA activation shows such non-linear behavior? It contradicts all previous known (to me) results. A discussion of this point would also be useful to the reader.

Our description and discussion of non-linear behaviour in our manuscript should only concern the relationship between the transcriptional input and the transcriptional output of a particular expression system. In the case of the CRISPRa device here, we would not treat the inducer concentration as the input because the relationship between the inducer concentration and the transcription rate of sgRNA under induction might already be sigmoidal / non-linear, which would carry forward and reflect on the graph of sfGFP level plotted against inducer concentration. Since we did not characterise the levels of sgRNA in that experiment, we could not convert the inducer concentrations into strengths of transcriptional inputs and could not plot transcriptional output (sfGFP level) versus transcriptional input (actual sgRNA level). Under this situation, we could not assess and compare the linearity or non-linearity of the system in a fair manner.

On the other hand, Figure 5d shows a two-tier cascade system. The reason why the input-output behaviour of the device with dxCas9 looks better is due to the output (sgRNA generated from the first layer) leakiness from the first layer decreased – which significantly lowered the basal output level of the second layer (see below schematic).

- Q15: Ok

- Q16: Please add this to the discussion, it will be useful for the readers.

We would be grateful if the reviewer could understand the length of this manuscript has already approached the limit, so we are honestly having difficulties to further incorporate discussion points without sacrificing the clarity on our findings. CRISPR regulation based cellular computing system could be an interesting research direction in the future. Any collaboration intention is welcome.

-- Q17: Ok

- Please use logarithmic scale in all fluorescence barplots wherever you have bars of small size (indistinguishable from zero, which is not informative at all).

We thank the Reviewer's suggestion. We used the linear axis here because the small size bars are generally negative control or data from 'OFF' state. For necessary figures (e.g. Supplementary Figure 13), we already employed logarithmic scale.

- Minor comments:

- Fig 1: the labelling of subpanels is awkward, I would use c, d, e instead of d, e, c

We have changed the order of subpanels as suggested.

- Page 4 and other places: replace "chimeric hexamers" by "heterohexamers " throughout the text as chimeric constructions are usually used for protein fusions.

We thank Reviewer's suggestion and we have reworded as suggested.

AJ

Reviewer #2 (Remarks to the Author):

N/A

References

1. Jovanovic, G., Rakonjac, J. & Model, P. In vivo and in vitro activities of the Escherichia coli sigma(54) transcription activator, PspF, and its DNA-binding mutant, PspF Delta HTH. *Journal of Molecular Biology* **285**, 469-483 (1999).
2. Dworkin, J., Jovanovic, G. & Model, P. Role of upstream activation sequences and integration host factor in transcriptional activation by the constitutively active prokaryotic enhancer-binding protein PspF. *Journal of Molecular Biology* **273**, 377-388 (1997).